# H3K9me1/2 methylation limits the lifespan of *daf-2* mutants in *C. elegans*

Meng Huang[1†], Minjie Hong[1†], Xinhao Hou[1], Chengming Zhu[1], Di Chen[2]*, Xiangyang Chen[1]*, Shouhong Guang[1,3]*, Xuezhu Feng[1]*

[1]The USTC RNA Institute, Ministry of Education Key Laboratory for Membraneless Organelles & Cellular Dynamics, Department of Obstetrics and Gynecology, The First Affiliated Hospital of USTC, School of Life Sciences, Division of Life Sciences and Medicine, Biomedical Sciences and Health Laboratory of Anhui Province, University of Science and Technology of China, Hefei, China; [2]State Key Laboratory of Pharmaceutical Biotechnology and MOE Key Laboratory of Model Animals for Disease Study, Model Animal Research Center, Institute for Brain Sciences, Nanjing University, Nanjing, China; [3]CAS Center for Excellence in Molecular Cell Science, Chinese Academy of Sciences, Hefei, China

*For correspondence:
chendi@nju.edu.cn (DC);
xychen91@ustc.edu.cn (XC);
sguang@ustc.edu.cn (SG);
fengxz@ustc.edu.cn (XF)

[†]These authors contributed equally to this work

**Abstract** Histone methylation plays crucial roles in the development, gene regulation, and maintenance of stem cell pluripotency in mammals. Recent work shows that histone methylation is associated with aging, yet the underlying mechanism remains unclear. In this work, we identified a class of putative histone 3 lysine 9 mono/dimethyltransferase genes (*met-2*, *set-6*, *set-19*, *set-20*, *set-21*, *set-32*, and *set-33*), mutations in which induce synergistic lifespan extension in the long-lived DAF-2 (insulin growth factor 1 [IGF-1] receptor) mutant in *Caenorhabditis elegans*. These putative histone methyltransferase plus *daf-2* double mutants not only exhibited an average lifespan nearly three times that of wild-type animals and a maximal lifespan of approximately 100 days, but also significantly increased resistance to oxidative and heat stress. Synergistic lifespan extension depends on the transcription factor DAF-16 (FOXO). mRNA-seq experiments revealed that the mRNA levels of DAF-16 Class I genes, which are activated by DAF-16, were further elevated in the *daf-2;set* double mutants. Among these genes, *tts-1*, *F35E8.7*, *ins-35*, *nhr-62*, *sod-3*, *asm-2*, and *Y39G8B.7* are required for the lifespan extension of the *daf-2;set-21* double mutant. In addition, treating *daf-2* animals with the H3K9me1/2 methyltransferase G9a inhibitor also extends lifespan and increases stress resistance. Therefore, investigation of DAF-2 and H3K9me1/2 deficiency-mediated synergistic longevity will contribute to a better understanding of the molecular mechanisms of aging and therapeutic applications.

## Editor's evaluation

This work presents important findings which pertain to how the length of an organism's lifespan can be affected by the presence of post-translational histone modifications, namely histone H3 lysine 9 mono- or di-methylation (H3K9me1/2). The authors find that the presence or absence of these marks plays a key role in modulating lifespan and stress resistance that synergizes with a long-lived insulin-pathway (daf-2) model. This work adds an important layer to aging regulation in organisms where a repressive chromatin state in specific genomic regions may limit the extent of lifespan. Hence, proper elucidation of this mechanism would be valuable to attain a comprehensive understanding of aging control and the reasons why these limits exist.

## Introduction

Lifespan is governed by complex interactions between genetic and environmental factors. The perturbation of insulin/insulin-like signaling, target of rapamycin (TOR) pathway, and mitochondrial functions have been shown to extensively modulate lifespan and health (*Fontana et al., 2010*; *Kenyon, 2010*; *López-Otín et al., 2013*). These genetic manipulations often lead to significant changes in gene expression at both the transcriptional and translational levels. Inhibition of DAF-2, the *Caenorhabditis elegans* ortholog of the insulin growth factor 1 (IGF-1) receptor, doubles adult lifespan by activating the DAF-16 (FOXO) transcription factor to regulate downstream genes involved in stress resistance, detoxification, and metabolism (*Kenyon et al., 1993*; *Kimura et al., 1997*; *Lin et al., 1997*; *Ogg et al., 1997*; *Murphy et al., 2003*; *McElwee et al., 2004*).

In addition to genetic regulation, aging is also modulated by epigenetic processes. Epigenetic marks, including histone acetylation and methylation, as well as the associated chromatin states, are altered during aging. Aging-dependent loss of chromatin repression is correlated with alterations in gene expression patterns, which have been documented in species ranging from *C. elegans* to humans (*Lee et al., 2000*; *Lund et al., 2002*; *Bennett-Baker et al., 2003*; *Lu et al., 2004*). For example, the methylation of histone 3 lysine 4 methylation (H3K4me) is one of the posttranslational histone modifications that marks on regions of active transcription (*Shilatifard, 2008*). H3K4me is deposited by the MLL/COMPASS complex, which travels with elongating RNA Polymerase II during transcription (*Wood et al., 2007*). In *C. elegans*, animals with reductions in COMPASS complex subunits (*wdr-5*, *ash-2*, and *set-2*) live longer than wild-type individuals (*Greer et al., 2010*).

However, little is known about the importance of repressive histone modification for aging marks. The H3K27me3 demethylase UTX-1 regulates lifespan independently of the presence of the germline but in a manner that depends on the insulin-FOXO signaling pathway (*Maures et al., 2011*). MET-2, a mammalian H3K9 methyltransferase SETDB1 homolog (*Loyola et al., 2006*), likely monomethylates and dimethylates H3K9 in *C. elegans* (*Towbin et al., 2012*). MET-2 is necessary both for a normal lifespan (*Tian et al., 2016*) and for the lifespan extension of *wdr-5* mutants (*Lee et al., 2019*). SET-6 is a putative H3K9me2/3 methyltransferase but not H3K9me1. Although the depletion of SET-6 does not significantly increase the lifespan of *C. elegans*, it may increase healthy aging (*Yuan et al., 2020*). SET-25 is a putative trimethylase for H3K9 (*Towbin et al., 2012*; *Padeken et al., 2021*). However, the deletion of SET-25 does not significantly change the worm lifespan (*Woodhouse et al., 2018*).

The enzymes responsible for histone lysine methylation are called histone methyltransferases (HMTs). HMTs typically contain a conserved catalytic domain called SET, which stems from **S**u(var)3–9, **E**nhancer of zeste, and **T**rithorax, the first HMTs known to carry this domain (*Tschiersch et al., 1994*). The *C. elegans* genome encodes 38 SET domain-containing proteins, of which 5 are essential for viability (*Andersen and Horvitz, 2007*; *Ni et al., 2012*). However, the biological roles of most SET proteins are largely unknown.

To investigate the function of repressive histone methylation in lifespan regulation, we selected a number of putative H3K9 methyltransferases as well as methyltransferases that may indirectly regulate H3K9me and tested whether the loss-of-function of these SET proteins could change the lifespan of *C. elegans*. Interestingly, we found that the H3K9me1/2, but not H3K9me3, mutants exhibited a synergistic lifespan extension with *daf-2* mutation. These animals showed an average lifespan of approximately 60 days, which is approximately 70% longer than that of *daf-2* worms and is three times as long as that of wild-type N2 animals. The double mutants exhibited a maximal lifespan of approximately 100 days. The synergistic lifespan extension of DAF-2 and lifespan-limiting HMT mutants depend on the transcription factor DAF-16 (FOXO). mRNA-seq experiments showed that DAF-16 Class I genes (genes that are activated by DAF-16) are activated in long-lived worms. Therefore, we conclude that H3K9me1/2 may limit the lifespan of *daf-2* animals by repressing the expression of DAF-16 Class I genes.

## Results

### The depletion of *set-21* extends lifespan and enhances stress resistance in *daf-2* mutant

SET-21 is a homolog of human isoform 1 of histone-lysine *N*-methyltransferase SUV39H2, with functions unknown in *C. elegans* (*Wenzel et al., 2011*). We accidentally found that *set-21(ok2320)*

significantly extended lifespan in *daf-2* mutants, but did not change the lifespan of N2 animals (*Figure 1A*, *Supplementary file 1* Statistical analyses of lifespan experiments). To confirm the function of SET-21 in lifespan regulation, we first generated a number of additional deletion alleles of *set-21* by CRISPR/Cas9 technology (*Figure 1—figure supplement 1A*). The deletion of *set-21* did not significantly extend the lifespan or change the brood size in N2 background (*Figure 1A*, *Figure 1—figure supplement 1B*). Strikingly, in *daf-2(e1370)* mutant worms, knocking out *set-21* reduced the brood size and significantly extended lifespan (*Figure 1A*, *Figure 1—figure supplement 1B*). The average lifespan of *daf-2(e1370);set-21(ust68)* were 55% longer than that of *daf-2(e1370)* animals (*Figure 1A*). And the maximal lifespan of *daf-2;set-21* animals achieved approximately 100 days, which is three times as long as that of N2 animals.

*eat-2* mutant is a genetic model in dietary restriction (DR) research in *C. elegans*. The mutation of *eat-2* renders a non-efficient pharynx in grinding bacteria and results in DR on regular media plates (*Raizen et al., 1995*). *eat-2* animals exhibit phenotypes similar to those observed in other species subjected to DR, including an ~36% longer lifespan (*Figure 1B*; *Lakowski and Hekimi, 1998*). Similarly, knocking out *set-21* further extended the lifespan of *eat-2(ad465)* mutant worms (*Figure 1B*). The average lifespan of *eat-2(ad465);set-21(ust68)* were 16% longer than that of *eat-2(ad465)* animals, suggesting a broader effect of these lifespan-limiting SET proteins.

The extended lifespan of nematodes has been shown to correlate with the activation of stress response genes and increased stress resistance (*Lithgow et al., 1995*; *Murphy et al., 2003*; *Samuelson et al., 2007*). The deletion of *set-21* did not significantly change the stress resistance in N2 background animals (*Figure 1C–D*). The *daf-2* mutation led to a higher resistance to oxidative stress and heat-shock stress than N2 animals. Strikingly, *daf-2;set-21* worms revealed a much higher resistance to oxidative stress via hydrogen peroxide treatment and heat-shock stress than *daf-2* animals (*Figure 1C–D*).

## The depletion of *met-2* extends lifespan in *daf-2* mutant worms and enhances stress resistance

The remarkable lifespan extension in *daf-2;set-21* animals inspired us to re-investigate the role of histone H3 lysine 9 methylation in *C. elegans*. MET-2 is the mammalian SETDB1 homolog, which is involved in mono- and dimethylation of H3K9 (*Figure 2A*; *Loyola et al., 2006*; *Towbin et al., 2012*). The brood size of *met-2* mutant is approximately half of that of wild-type N2 animals (*Figure 2B*). Interestingly, *daf-2;met-2* double mutant exhibited synthetic defect and had a significantly reduced brood size. *met-2* mutants exhibited a modest shorter lifespan than wild-type N2 animals (*Figure 2C*; *Tian et al., 2016*). Strikingly, *daf-2;met-2* double mutants revealed an average lifespan of approximately 47 days, which is 30% longer than that of *daf-2* mutation alone and is 2.3 times as long as that of wild-type N2 animals (*Figure 2C*). The depletion of *met-2* enhanced the oxidative stress resistance and heat stress resistance in both N2 and *daf-2* mutant worms (*Figure 2D–E*). These results suggest that *met-2* and H3K9me1/2 may play different roles in lifespan regulation in N2 and *daf-2* animals. Alternatively, the depletion of *daf-2* may provide a sensitized genetic background to identify genes that are involved in aging and lifespan regulation.

SET-25 is a putative trimethylase for H3K9 (*Figure 2A*; *Towbin et al., 2012*). However, deletion of SET-25 did not significantly change the worm lifespan or stress resistance in either the wild-type N2 or *daf-2* background animals (*Figure 2C–E*). The deletion of SET-25 only moderately enhanced the oxidative stress resistance in *daf-2* mutant worms (*Figure 2D–E*).

## Identification of *set* genes required for lifespan limitation in *daf-2* animals

*C. elegans'* genome encodes 38 proteins containing SET domains, most of which have unknown function (*Andersen and Horvitz, 2007*). To investigate the function of H3K9 methylation in lifespan regulation, we searched for other SET proteins that exhibit sequence similarity to SET-21 (*Figure 3A*, *Figure 3—figure supplement 1*). We selected seven additional SET proteins, including SET-6, SET-13, SET-15, SET-19, SET-20, SET-32, and SET-33, which are most closely related to the SET-21 in the dendrogram. All these proteins contain conserved catalytic SET domains and are annotated as putative H3K9 methyltransferases, as well as methyltransferases that may indirectly regulate H3K9me. Among them, SET-6 is required for H3K9me2/3 methylation (*Yuan et al., 2020*) and SET-32 is required

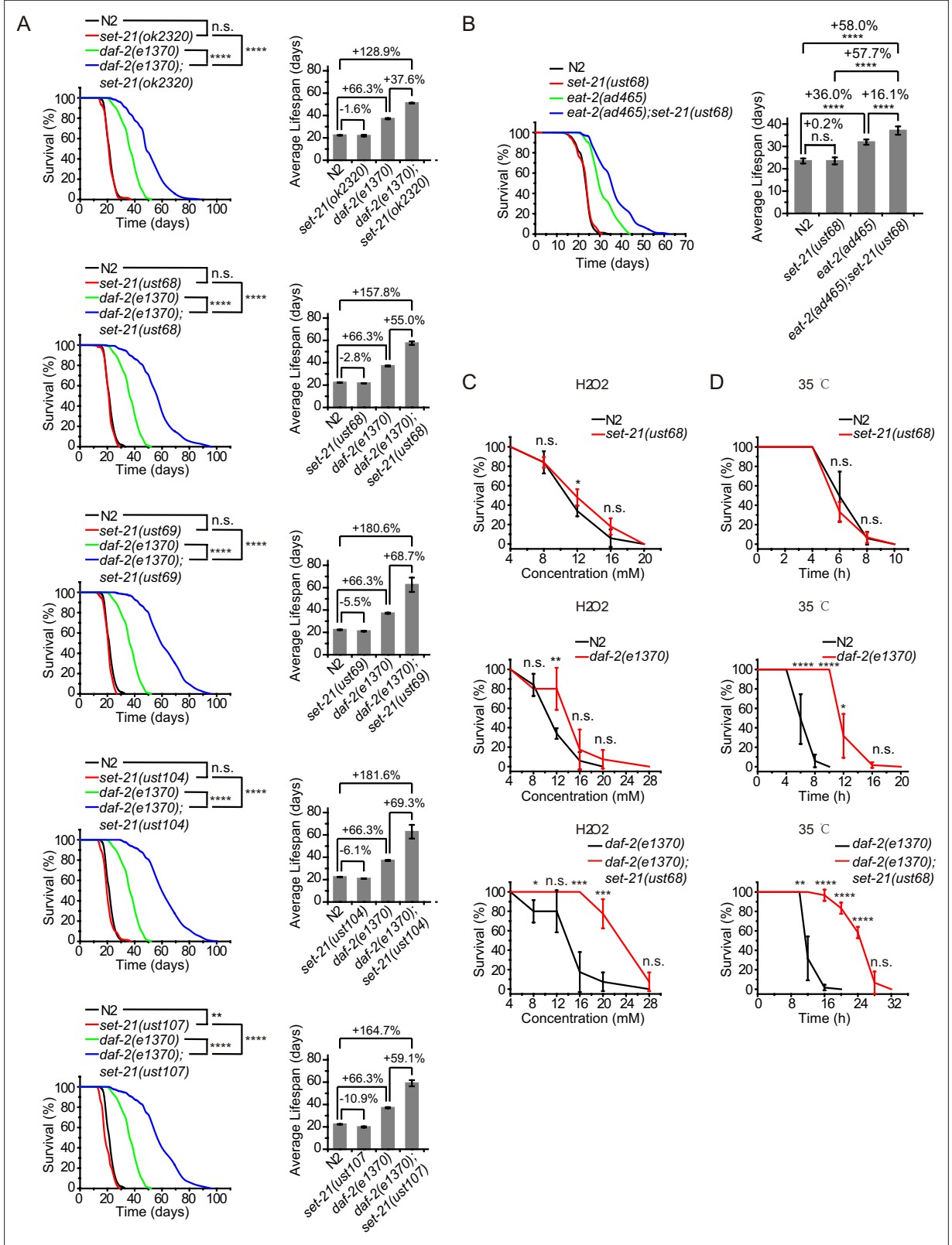

**Figure 1.** Synergistic lifespan extension and stress resistance in *daf-2;set-21* mutants. (**A**) (Left) Survival curves of indicated animals. (Right) Histogram displaying the average lifespan of the indicated animals. Mean ± s.e.m. of two independent experiments. The percentage of change was compared to the average lifespan of control animals. Asterisks indicate significant differences using log rank tests. ** 0.001< p < 0.01; ****p<0.0001. n.s., not significant, p>0.05. Lifespan data were summarized in *Supplementary file 1*. (**B**) (Left) Survival curves of indicated animals. (Right) Histogram displaying

*Figure 1 continued on next page*

*Figure 1 continued*

the average lifespan of the indicated animals. Mean ± s.e.m. of two independent experiments. Asterisks indicate significant differences using log rank tests. ****p<0.0001. Lifespan data were summarized in **Supplementary file 1**. (**C, D**) Survival curves of the indicated animals. (**C**) Oxidative and (**D**) heat stress. Data in each panel are presented as the mean ± s.e.m. of five independent experiments. Asterisks indicate significant differences using two-tailed t tests. *p<0.05; **p<0.01; ***p<0.001; ****p<0.0001; n.s., not significant, p>0.05. The same oxidative stress data of control N2 and *daf-2* animals were used in (**C**) **Figures 2 and 3** and **Figure 3—figure supplement 3A**. The same heat stress data of control N2 and *daf-2* animals were used in (**D**) **Figures 2 and 3** and **Figure 3—figure supplement 3B**.

The online version of this article includes the following figure supplement(s) for figure 1:

**Figure supplement 1.** Alleles of *set-21*.

for nuclear RNAi-directed H3K9me3 in *C. elegans* (**Mao et al., 2015**; **Kalinava et al., 2017**; **Spracklin et al., 2017**). Using N2 and the sensitized *daf-2* background, we assayed the lifespan of these putative H3K9 methyltransferase mutants.

We acquired additional *set* mutants from CGC and also generated alleles by CRISPR/Cas9 technology (**Figure 3—figure supplement 2A**). Among them, the deletion of *set-6, set-19, set-20,* and *set-32* modestly increased the lifespan compared to that of wild-type N2 animals (**Figure 3B**). However, in the *daf-2* mutant background, mutations of *set-6, set-19, set-20, set-32,* and *set-33* exhibited a striking synergistic lifespan extension. While the lifespan of *daf-2;set-20* and *daf-2;set-32* are approximately 60% longer than that of *daf-2* worms, the *daf-2;set-6* and *daf-2;set-19* live 70% longer than that of *daf-2* animals (**Figure 3C**). Especially, *daf-2;set-19* exhibits a maximal lifespan of approximately 100 days, which is five times as long as the average lifespan of N2 animals. *daf-2;set-6, daf-2;set-19, daf-2;set-20, daf-2;set-32, daf-2;set-33* had fewer brood sizes than the single *set* gene mutants (**Figure 3—figure supplement 2B**). The long-lived animals were also much more resistant than control animals to the oxidative stress induced by hydrogen peroxide (**Figure 3D**, **Figure 3—figure supplement 3A**) and heat stress (**Figure 3E**, **Figure 3—figure supplement 3B**) in *daf-2* mutant worms but not in N2 background animals. *set-32(ok1457)* revealed a higher resistance to heat stress but not oxidative stress than N2 animals.

To test whether SET-6, SET-19, SET-20, SET-21, SET-32, and SET-33 act in the same genetic pathway to regulate lifespan in *daf-2* animals, we crossed *set-21(ust68)* to other *set* mutants. As expected, the triple mutants *daf-2;set-21;set-6, daf-2;set-21;set-19, daf-2;set-21;set-20, daf-2;set-21;set-32,* and *daf-2;set-21;set-33* did not significantly further extend the lifespan than the double mutants (**Figure 4A**). Thus, these *set* genes probably act in the same genetic pathway to regulate lifespan.

We speculated that there might be a correlation between the expression pattern and lifespan extension of the *set* genes Thus, we generated in situ single-copy 3xFLAG-GFP tagged SET-21, SET-25, and SET-32 transgenes by CRISPR/Cas9 technology and single-copy GFP-3xFlAG-tagged SET-13 transgenic strains using the Mos1-mediated single-copy insertion (MosSCI) technology. These animals were imaged in N2 and *daf-2* background at embryonic and larva stages, respectively, to figure out the expression patterns and cellular localizations of SET proteins (**Figure 4—figure supplement 1**). However, we failed to find a clear correlation between their expression pattern with the lifespan extension phenotype. The depletion of SET-21 and SET-32 extended the lifespan of *daf-2* animals, but GFP::SET-21 and GFP::SET-32 showed different expression patterns and subcellular localizations. GFP::SET-21 exclusively located in the nucleus but GFP::SET-32 located in both the nucleus and cytoplasm in embryos. At larval stage, GFP::SET-21 was expressed in soma but GFP::SET-32 was expressed in germline. Both *set-13* and *set-25* mutants did not show lifespan extension in N2 and *daf-2* background worms. SET-13 and SET-25 were enriched in the nucleus. SET-13::GFP was expressed through eight-cell stage to late-embryonic stage, while SET-25::GFP was expressed in the whole embryonic stage and the germline in young adults.

H3K9me3 is a hallmark of heterochromatin and is often enriched on repetitive sequences (**Peters et al., 2003**; **Guenatri et al., 2004**; **Martens et al., 2005**). The heterochromatin protein 1 (HP1) is a highly conserved small nonhistone protein, which was first identified in *Drosophila*, and recognizes H3K9me3 (**James and Elgin, 1986**). *C. elegans* has two HP1 paralogs: HP1 Like 1 and 2 (HPL-1 and HPL-2) (**Couteau et al., 2002**). The chromatin binding pattern of HPL-2 most closely resembles H3K9me1 and H3K9me2 and less closely resembles H3K9me3 (**Garrigues et al., 2015**). *hpl-2(tm1489)* mutation has been shown to extend lifespan (**Meister et al., 2011**). However, both *daf-2;hpl-1* and *daf-2;hpl-2* double mutants did not exhibit further lifespan extension compared to the *daf-2* mutant

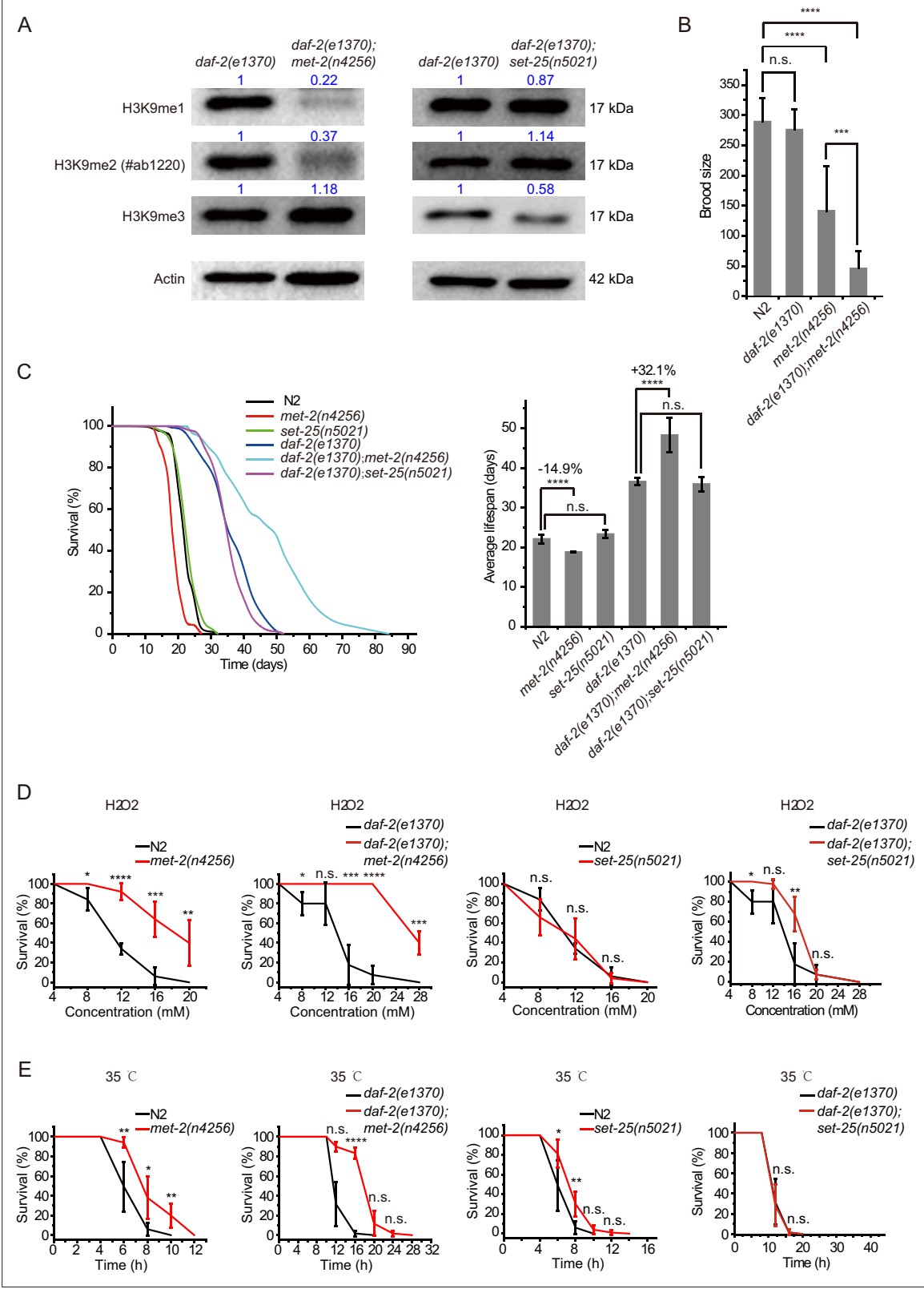

**Figure 2.** *daf-2;met-2* mutants revealed extended lifespan and increased resistance to oxidative and heat stress compared to *daf-2* animals. (**A**) Western blotting of L4 stage animals with the indicated antibodies. Numbers indicate the scanned density by ImageJ. (**B**) Brood size of indicated animals at 20°C. (**C**) (Left) Survival curves of indicated animals. (Right) Histogram displaying the average lifespan of the indicated animals. Mean ± s.e.m. of three independent experiments. Asterisks indicate significant differences using log rank tests. ****p<0.0001. Lifespan data were summarized in

*Figure 2 continued on next page*

*Figure 2 continued*

**Supplementary file 1**. (**D, E**) Survival curves of the indicated animals. (**D**) Oxidative and (**E**) heat stress. Data in each panel are presented as the mean ± s.e.m. of five independent experiments. Asterisks indicate significant differences using two-tailed t tests. *p<0.05; **p<0.01; ***p<0.001; ****p<0.0001; n.s., not significant, p>0.05.

The online version of this article includes the following source data for figure 2:

**Source data 1.** Original files and figures of the blots in *Figure 2A*.

(*Figure 4B*). SET-25 colocalized with both H3K9me3 and HPL-1, but did not colocalize with HPL-2 (*Towbin et al., 2012*). SET-25 is required for H3K9me3 methylation but is not involved in lifespan limitation (*Figure 2A and C*), further supporting that H3K9me3 may be dispensable for lifespan regulation in *daf-2* animals.

Previous work has shown that knocking out the H3K4me3 methyltransferase SET-2 extended worm lifespan of N2 animals (*Greer et al., 2010*). To test whether SET-2-dependent H3K4me3 and lifespan-limiting H3K9 methyltransferases act in the same genetic pathway to regulate lifespan, we crossed *set-2* to *daf-2;set-21* animals. The double mutants *daf-2;set-2* and *daf-2;set-21* and the triple mutants *daf-2;set-2;set-21* exhibited similar lifespan extensions relative to that of *daf-2* animals (*Figure 4C*), suggesting that H3K4me3 and H3K9me1/2 may function in the same genetic pathway or regulate the same cohorts of target genes for lifespan modulation.

Inhibition of RSKS-1 (S6K), the TOR pathways, extends lifespan in *C. elegans* (*Hansen et al., 2007*; *Pan et al., 2007*), and the double mutant of *daf-2;rsks-1* leads to synergistically prolonged longevity (*Chen et al., 2013*). However, the triple mutants *daf-2(e1370);rsks-1(ok1255);set-21(ust68)* revealed similar lifespan as long as those of *daf-2(e1370);rsks-1(ok1255)* and *daf-2(e1370);set-21(ust68)* double mutants (*Figure 4D*), suggesting that either *rsks-1* and *set-21* act in the same molecular pathway to regulate lifespan or there is likely an upceiling of maximal lifespan of approximately 100 days for *C. elegans*. Interesting, a report suggested that in a mutant of *age-1*, which encodes the class-I phosphatidylinositol 3-kinase catalytic subunit (PI3K(CS)), worms can survive to a median of 145–190 days at 20 degrees and a maximal lifespan of approximately 260 days, with nearly 10-fold extension of both median and maximum adult lifespan relative to control animals (*Ayyadevara et al., 2008*).

Therefore, we concluded that a number of *set* genes limit the lifespan and stress resistance of *daf-2* animals in *C. elegans*.

## Lifespan-limiting SET proteins regulate DAF-16 Class I genes in *daf-2* animals

The longevity phenotype of *daf-2* animals depends on the downstream DAF-16 transcription factor (*Kenyon et al., 1993*). DAF-16 localizes in the cytoplasm in N2 worms but accumulates in the nucleus upon *daf-2* mutation (*Henderson and Johnson, 2001*; *Lin et al., 2001*). To determine whether the expression pattern and subcellular localization of DAF-16 are altered upon the mutation of these lifespan-limiting HMTs, we crossed a widely used high-copied DAF-16::GFP transgene with *daf-2*, *set-21*, *set-25*, *daf-2;set-21*, and *daf-2;set-25* animals. As reported, the mutation of *daf-2* resulted in a pronounced nuclear accumulation of DAF-16::GFP (*Figure 5—figure supplement 1A*). In N2 background, the mutation of *set-21* or *set-25* did not detectably change the cytoplasmic localization of DAF-16::GFP (*Figure 5—figure supplement 1A*). In *daf-2* background worms, the mutation of *set-21* or *set-25* did not further induce detectable changes in the expression pattern and subcellular localization of DAF-16::GFP.

We generated a single-copied DAF-16::GFP transgene by inserting a GFP tag onto the C-terminal of DAF-16 via CRISPR/Cas9 technology (*Figure 5A*). Then, we crossed the single-copied DAF-16::GFP transgene with *daf-2, set-21*, *set-25*, *daf-2;set-21*, and *daf-2;set-25* animals. DAF-16::GFP accumulated in the cytoplasm in N2 background animals. Unlike in the high-copied DAF-16::GFP transgenic animal, *daf-2* mutation only modestly enriched DAF-16::GFP to the nucleus in the single-copied DAF-16::GFP transgenic strain (*Figure 5B* and *Figure 5—figure supplement 1B*). Strikingly, in *daf-2;set-21* and *daf-2;set-25* double mutant animals, DAF-16::GFP was strongly enriched in the nucleus in the single-copied transgenic animals. The reason for the subcellular localization difference between the high-copied and single-copied DAF-16::GFP transgenic animals is unclear. However, the single-copied DAF-16::GFP may be used to re-visit the mechanism of lifespan regulation pathways. In addition,

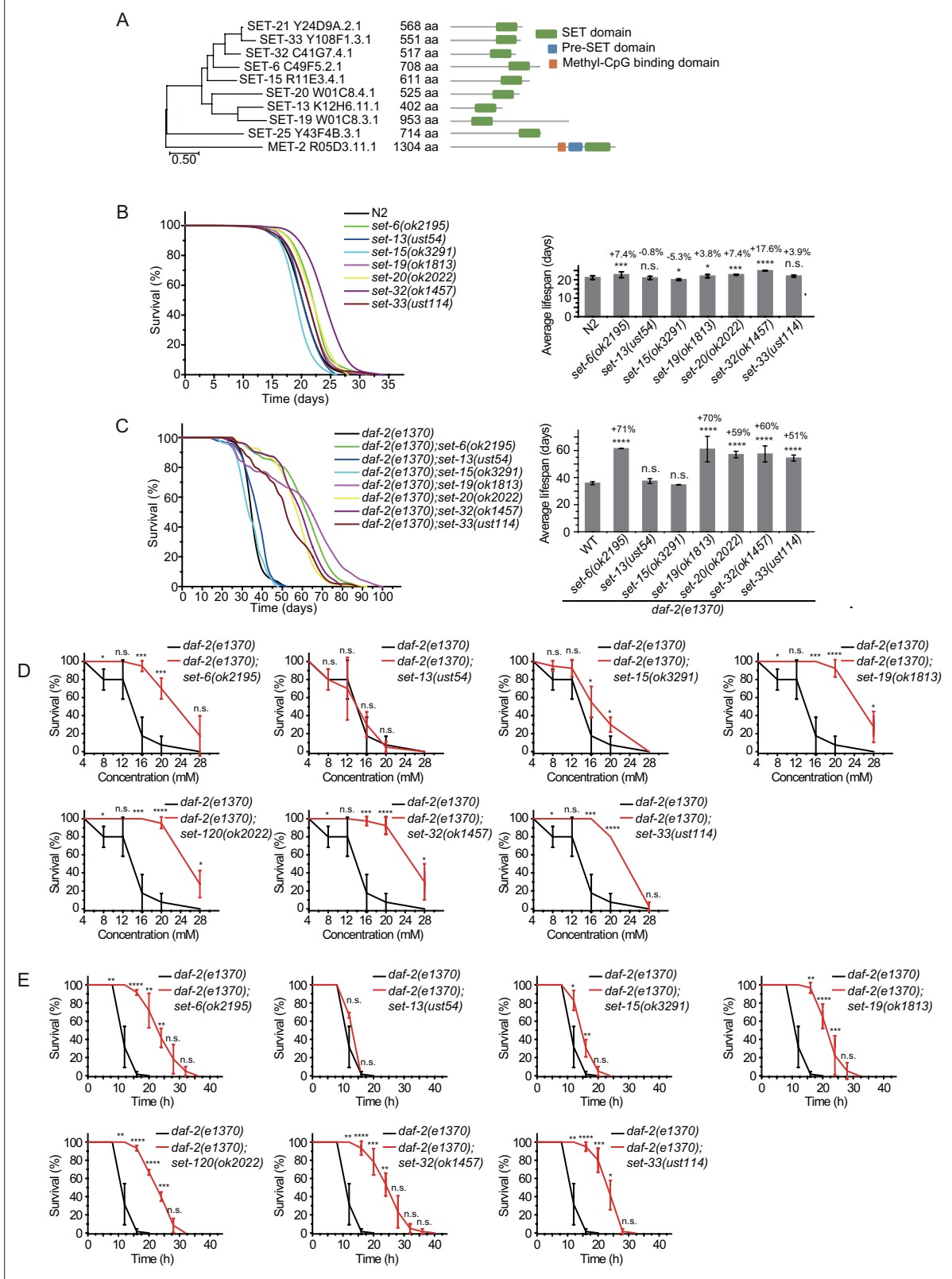

**Figure 3.** Synergistic lifespan extension of *set-6, set-19, set-20, set-32,* and *set-33* with *daf-2* animals. (**A**) Dendrogram comparing the protein sequences of SET proteins that are putative H3K9 methyltransferases or methyltransferases that may indirectly regulate H3K9me in *Caenorhabditis elegans*. The numbers indicate the length of each protein. (**B**) (Left) Survival curves of indicated animals. (Right) Histogram displaying the average lifespan of the indicated animals. Mean ± s.e.m. of two independent experiments. The percentage of change was compared to the average lifespan of N2 animals.

*Figure 3 continued on next page*

*Figure 3 continued*

Asterisks indicate significant differences using log rank tests. *p<0.05; **p<0.01; ***p<0.001; ****p<0.0001; n.s., not significant, p>0.05. Lifespan data were summarized in ***Supplementary file 1***. (**C**) (Left) Survival curves of indicated animals. (Right) Histogram displaying the average lifespan of the indicated animals. Mean ± s.e.m. of two independent experiments. The percentage of change was compared to the average lifespan of *daf-2* animals. Asterisks indicate significant differences using log rank tests. *p<0.05; **p<0.01; ***p<0.001; ****p<0.0001; n.s., not significant, p>0.05. (**D, E**) Survival curves of the indicated animals. (**D**) Oxidative and (**E**) heat stress. Data are presented as the mean ± s.e.m. of five independent experiments. Asterisks indicate significant differences using two-tailed t tests. *p<0.05; **p<0.01; ***p<0.001; ****p<0.0001; n.s., not significant, p>0.05.

The online version of this article includes the following figure supplement(s) for figure 3:

**Figure supplement 1.** Dendrogram comparing the protein sequences of all SET proteins in *Caenorhabditis elegans*.

**Figure supplement 2.** Alleles of *set* genes and the brood size of *set* mutants.

**Figure supplement 3.** The depletion of *set* genes did not induce significant stress resistance in N2 background animals.

the mechanism of enhanced nuclear accumulation of DAF-16::GFP in *daf-2;set-21* and *daf-2;set-25* animals is unknown. Given that *daf-2;set-21* exhibited extended lifespan but *daf-2;set-25* did not, and both of them revealed strong nuclear localization of DAF-16::GFP, it is quite possible that there is an additional level of regulation downstream of the nuclear localization of DAF-16.

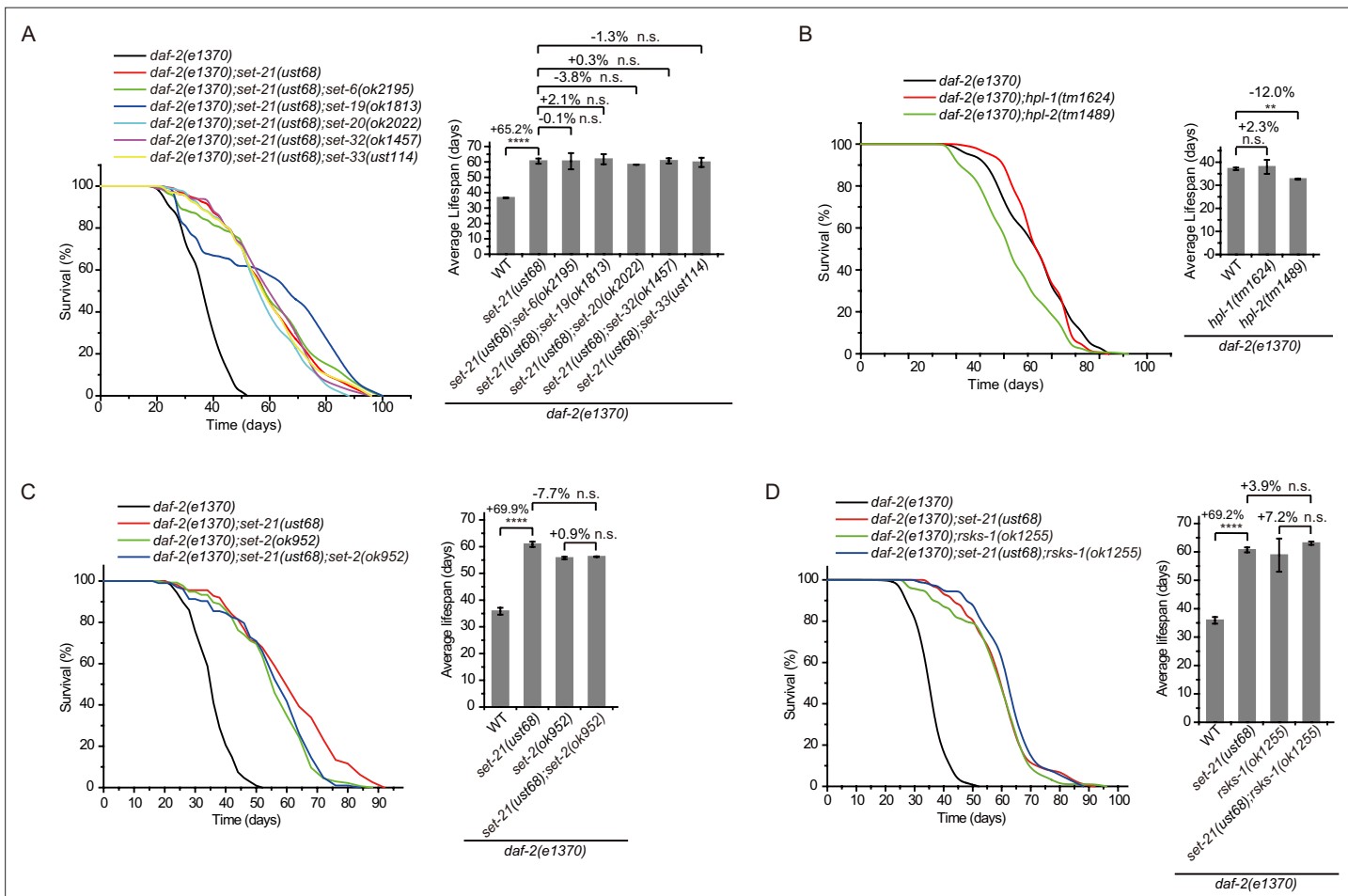

**Figure 4.** Genetic pathway analysis of *set* genes in lifespan regulation in *daf-2* mutants. (**A–D**) (Left) Survival curves of the indicated animals. (Right) Histogram displaying the average lifespan of the indicated animals. Means ± s.e.m. of two independent experiments. The percentage change was compared to the lifespan of control animals. Asterisks indicate significant differences using log rank tests. ** 0.001< p < 0.01; ****p<0.0001; n.s., not significant, p>0.05. Lifespan data were summarized in ***Supplementary file 1***.

The online version of this article includes the following figure supplement(s) for figure 4:

**Figure supplement 1.** Expression pattern of selected SET proteins.

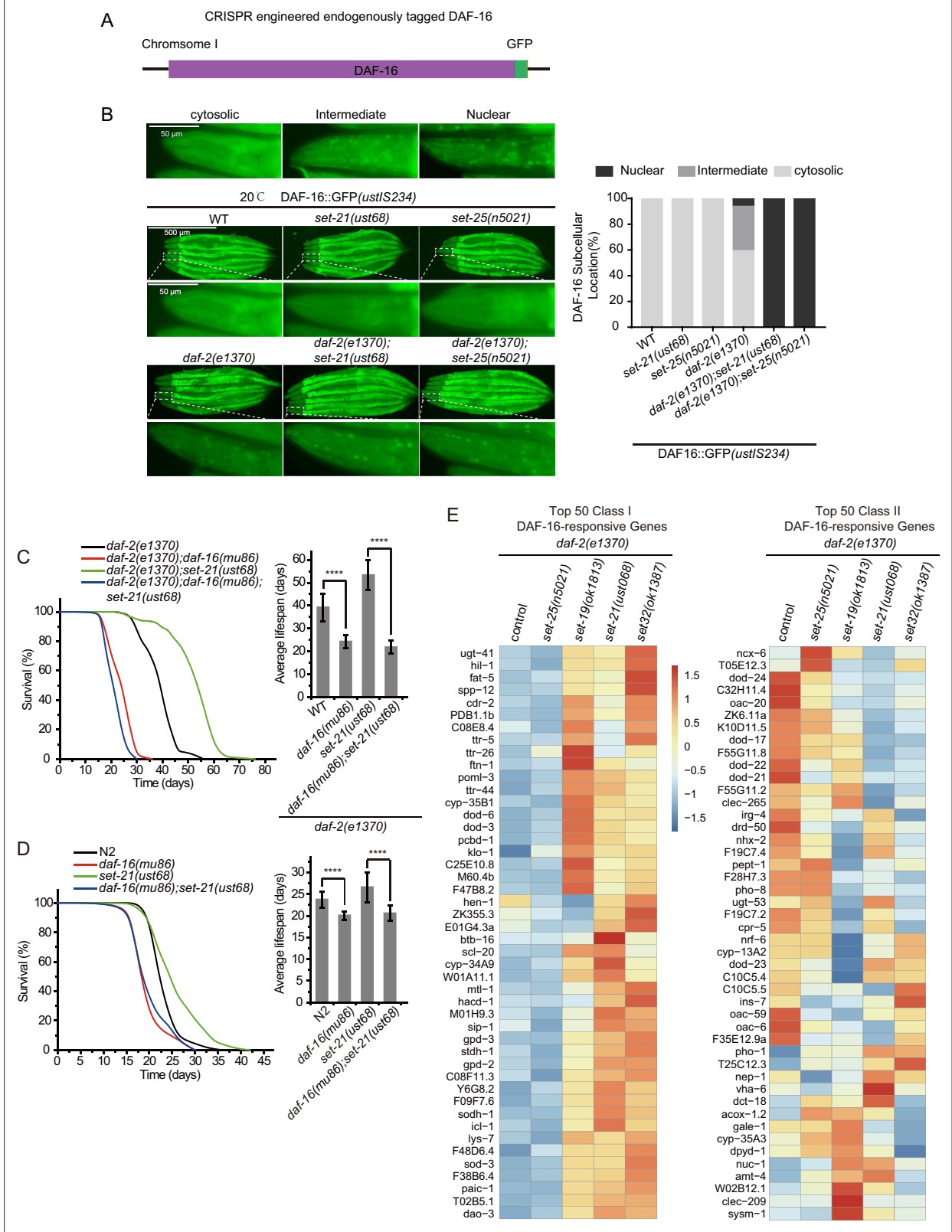

**Figure 5.** The lifespan-limiting SET proteins regulate DAF-16 Class I genes in *daf-2* mutants. (**A**) Construction of an in situ GFP-tagged DAF-16 transgene by CRISPR/Cas9 technology. The subcellular localization of DAF-16::GFP was scored at the pharynx in young adult animals. (**B**) (Left) Fluorescent images of the single-copied DAF-16::GFP transgene in indicated animals at 20°C. (Right) The subcellular localization of the single-copied DAF-16::GFP transgene was scored in indicated animals. (**C, D**) (Left) Survival curves of indicated animals. (Right) Histogram displaying the average

*Figure 5 continued on next page*

*Figure 5 continued*

lifespan of the indicated animals. Mean ± s.e.m. of three independent experiments. Asterisks indicate significant differences using log rank tests. ****p<0.0001. Lifespan data were summarized in **Supplementary file 1**. (**E**) Heatmap of the standardized fragments per kilobase of transcript per million mapped reads (FPKM) of reported top 50 DAF-16 Class I and Class II genes by mRNA-seq in the indicated animals. Statistical analysis was performed to obtain the expression levels of each gene in each strain and the average of the expression levels in the five strains. The gene expression levels in a single strain were compared with the average of the five strains, and the ratio obtained was processed by log2. The resulting value>0 means the expression level of the gene in the indicated strain is higher than the average expression levels of this gene in the five strains, as shown from yellow to red. The resulting value<0 means the expression level of the gene in the indicated strain is lower than the average expression levels of this gene in the five strains, as shown in blue. The expression levels are indicated by the color bar. The mRNA-seq was conducted in single replicate and validated by quantitative real-time PCR in **Figure 7A**. The 50 most significant positive (Class I) and negative (Class II) targets of DAF-16 were selected according to their responsiveness to DAF-16 (**Tepper et al., 2013**).

The online version of this article includes the following figure supplement(s) for figure 5:

**Figure supplement 1.** Fluorescent images of GFP-tagged DAF-16 in the indicated young adult animals.

**Figure supplement 2.** Class I DAF-16 genes were upregulated in *daf-2;set-19, daf-2;set-21,* and *daf-2;set-32* worms.

To further test the role of DAF-16 in the *daf-2;set-21*-induced synergistic longevity, we constructed a *daf-2(e1370);daf-16(mu86);set-21(ust68)* triple mutant. The *daf-16* mutation reverted the prolonged longevity phenotype of *daf-2;set-21* to an average lifespan of 23 days, which is similar to that of *set-21* or N2 alone (**Figure 5C**). In N2 background, *daf-16* and *daf-16;set-21* mutant worms lived shorter than wild-type N2 and *set-21* animals (**Figure 5D**).

The depletion of DAF-2 reduces the insulin signaling pathway and leads to both upregulation and downregulation of large sets of genes, referred to as DAF-16 Class I and II genes, respectively (**Murphy et al., 2003**). Class I genes are induced in *daf-2* mutants but are repressed in *daf-2;daf-16* double mutants. Class II genes are not induced in *daf-2* mutants but are induced in *daf-2; daf-16* double mutants; 1663 genes are classified as positive (Class I) DAF-16 targets and 1733 genes are classified as negative (Class II) DAF-16 targets of DAF-16 (**Tepper et al., 2013**). Class I genes are enriched for the Gene Ontology categories including oxidation, reduction, and energy metabolism, whereas Class II genes are enriched for genes involved in biosynthesis, growth, reproduction, and development. We performed mRNA-seq and analyzed Class I and II DAF-16 genes to identify the mis-regulated genes in *daf-2;set* mutants. Interestingly, the mRNA levels of DAF-16 Class I, but not Class II, genes are consistently activated in long-lived *daf-2;set-19, daf-2;set-21,* and *daf-2;set-32* worms, than in control *daf-2* and *daf-2;set-25* animals (**Figure 5E**, **Figure 5—figure supplement 2**).

## Identification of the lifespan-limiting SET targeted genes in *daf-2* animals

To identify target genes in this group of lifespan extension *daf-2;set* mutants, we re-analyzed the mRNA expression profile in the double mutants (**Figure 6**, **Figure 6—figure supplement 1**). We identified 49 co-upregulated genes and 11 co-downregulated genes that are specifically enriched in the long-lived *daf-2;set-19, daf-2,set-21, daf-2;set-32* double mutants, but not in short-lived control *daf-2* and *daf-2;daf-25* animals (**Figure 6A–B** and **Figure 6—figure supplement 1A**). Interestingly, among the 49 co-upregulated genes, 27 of them are also DAF-16 Class I genes (**Figure 6—figure supplement 1B**). Twenty-two co-upregulated genes are not DAF-16 Class I genes, suggesting the existence of additional regulations.

We then analyzed a number of known transcription factors for their binding to the co-regulated targeted genes (**Figure 6C–D**, **Figure 6—figure supplement 1C-E**). Among them, DAF-16 and NHR-80 were specifically enriched at the transcription start sites (TSSs) of the 49 co-upregulated genes. NHR-80 is a homolog of mammalian hepatocyte nuclear factor 4 and is an important nuclear hormone receptor involved in the control of fat consumption and fatty acid composition in *C. elegans* (**Goudeau et al., 2011**; **Pathare et al., 2012**). Among the genes targeted by DAF-16 and NHR-80, 12 of them are co-regulated by both factors, which is consistent with previous report that *daf-16* and *nhr-80* function in parallel pathway for lipid metabolism (**Wan et al., 2022**).

To confirm the change in target gene expression, we chose 10 genes from the 49 co-upregulated genes. Among them, *nhr-62, dao-3,* and *tts-1* are direct binding targets of both DAF-16 and NHR-80. *sod-3* is a DAF-16 direct binding target. *lys-7* and *asm-2* are NHR-80 direct binding targets. All of the 10 genes have been shown to be upregulated in *daf-2* compared to *daf-2;daf-16* animals (**Tepper**

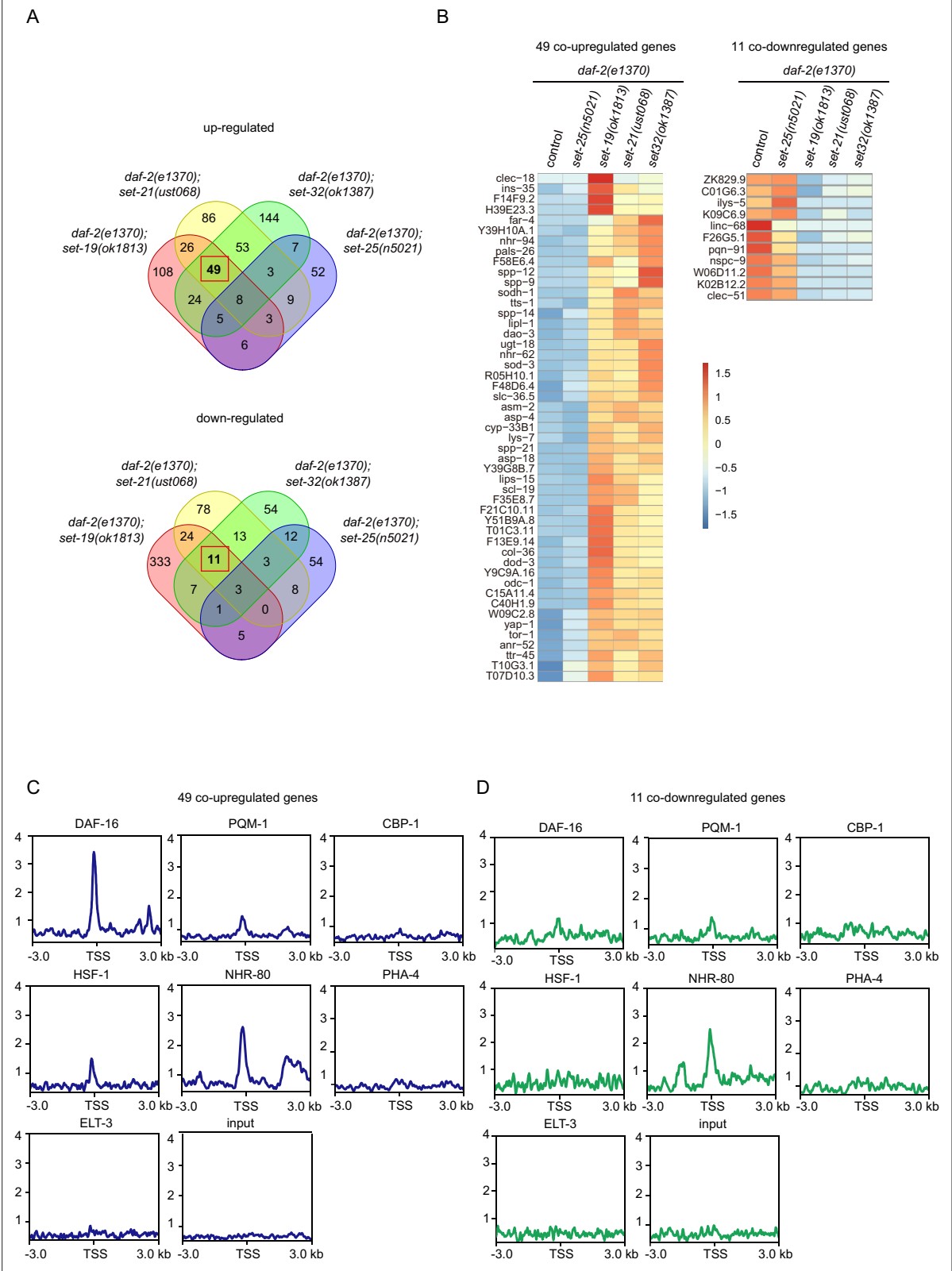

**Figure 6.** mRNA-seq identified differentially expressed genes co-regulated by the lifespan-limiting SET proteins in *daf-2* animals. (**A**) Venn diagrams showing the overlapped genes upregulated or downregulated between *daf-2(e1370);set-25(n5021)*, *daf-2(e1370);set-19(ok1813)*, *daf-2(e1370);set-21(ust068)*, *daf-2(e1370);set-32(ok1387)* compared with *daf-2(e1370)*. Upregulated genes were defined as fold change ≥ 2and p<0.05. Downregulated genes were defined as fold change ≤ 0.5and p<0.05. (**B**) Heatmap of the standardized fragments per kilobase of transcript per million mapped reads

*Figure 6 continued on next page*

*Figure 6 continued*

(FPKM) for the 49 co-upregulated genes and 11 co-downregulated genes in the indicated animals. The expression levels are indicated by the color bar. (**C, D**) Profile plots showing the chromatin immunoprecipitation sequencing (ChIP-seq) signals of seven transcription factors around transcription start sites (TSSs) of (**C**) 49 co-upregulated genes and (**D**) 11 co-downregulated genes. The ChIP-seq datasets were downloaded from the ENCODE or NCBI GEO databases (***Supplementary file 6***).

The online version of this article includes the following figure supplement(s) for figure 6:

**Figure supplement 1.** The co-upregulated genes by the lifespan-limiting SET proteins are targeted by DAF-16 and NHR-80.

---

*et al., 2013*; *Essers et al., 2015*). We quantified the mRNA levels by quantitative real-time PCR (qRT-PCR) in *daf-2*, *daf-2;set-21* and *daf-2;set-25* animals (***Figure 7A***). Consistently, all the 10 genes were upregulated in *daf-2;set-21* double mutants than in *daf-2* and *daf-2;set-25* animals (***Figure 7A***).

We generated deletion mutants of these genes by CRISPR/Cas9 technology (***Figure 7—figure supplement 1***) and crossed the mutants into *daf-2;set-21* background. Seven of these mutants, *tts-1*, *nhr-62*, *ins-35*, *sod-3*, *asm-2*, *F35E8.7*, and *Y39G8B.7*, could partially shorten the lifespan extension phenotype of *daf-2;set-21* double mutant animals, but not shorten the lifespan of *daf-2* mutants (***Figure 7B***). Among them, *tts-1* is a ribosome-associated long noncoding RNA in *daf-2* mutant. Depleting *tts-1* in *daf-2* mutants has been reported to increase ribosome levels and shorten the extended lifespan (***Tepper et al., 2013***; ***Essers et al., 2015***). NHR-62 is a nuclear hormone receptor with DNA binding activity, which is required for DR-induced longevity in *C. elegans* (***Heestand et al., 2013***). INS-35 is an insulin-like peptides and may act on DAF-2 to regulate UPR^mt and mitochondrial dynamics (***Chen et al., 2021***). SOD-3 is a superoxide dismutase that is involved in the removal of superoxide radicals and required for lifespan extension in *isp-1* mutant worms (***Dues et al., 2017***). ASM-2 is an ortholog of human SMPD1 (sphingomyelin phosphodiesterase 1) and is involved in ceramide biosynthetic processes and sphingomyelin catabolic processes (***Lin et al., 1998***; ***Deng et al., 2008***). Strikingly, both F35E8.7 and Y39G8B.7 are predicted to encode proteins with ShK domain-like or ShKT domains, yet their functions are unknown.

Therefore, we concluded that the deletion of lifespan-limiting *set* genes in *daf-2* mutants leads to a synergistically extended lifespan by increasing DAF-16 activity.

## Lifespan-limiting SET proteins are required for H3K9me1/2 modification in *daf-2* animals

To further investigate the mechanism by which these SET proteins limit *daf-2* mutants' lifespan, we used Western blotting assay to examine the levels of a number of histone 3 methylation marks in L4 animals and embryos. Previous work have reported that SET-6 is required for H3K9me2/3 methylation (***Yuan et al., 2020***), SET-25 is required for H3K9me3 methylation (***Figure 2A***; ***Towbin et al., 2012***), and SET-32 is required for H3K23 methylation (***Schwartz-Orbach et al., 2020***).

The mutation of *daf-2* did not significantly change the levels of H3K9me1/2/3 (***Figure 8—figure supplement 1A***). Similarly, the mutations of the *set* genes also did not significantly change the levels of tested histone modifications compared to those in wild-type N2 animals, except that *set-19* was required for the accumulation of H3K23me3 in both embryos and L4 staged animals (***Figure 8—figure supplement 1B-C***). Interestingly, in the *daf-2* mutant background, the long-lived *daf-2;set-6*, *daf-2;set-19*, *daf-2;set-20*, *daf-2;set-21*, *daf-2;set-32*, and *daf-2;set-33* mutants, but not the short-lived *daf-2;set-13*, *daf-2;set-15*, and *daf-2;set-25* animals, decreased global H3K9me1/2 levels at the L4 larval stage (***Figure 8A***). We used another anti-H3K9me2 antibody #ab176882 and confirmed that *daf-2;set-6*, *daf-2;set-19*, *daf-2;set-20*, *daf-2;set-21*, *daf-2;set-32*, and *daf-2;set-33* mutants, but not *daf-2;set-13*, *daf-2;set-15* and *daf-2;set-25* animals, decreased global H3K9me1/2 levels at the L4 larval stage (***Figure 8—figure supplement 2A***). In embryos, *daf-2;set-15*, *daf-2;set-20*, and *daf-2;set-32* mutants reduced H3K9me1/2 levels (***Figure 8—figure supplement 2B***). The *daf-2;set-19* mutant also showed decreased H3K23me3 levels (***Figure 8—figure supplement 2B***). Although H3K4me has been shown to be involved in lifespan regulation, none of the long-lived *daf-2;set-6*, *daf-2;set-19*, *daf-2;set-20*, *daf-2;set-21*, *daf-2;set-32*, and *daf-2;set-33* mutants revealed significant changes in H3K4 methylation levels (***Figure 8A***, ***Figure 8—figure supplement 2B***). Chromatin immunoprecipitation (ChIP) followed by qRT-PCR further revealed a modest reduction in H3K9me1/2 levels of the 10 target genes in *daf-2;set-21* mutants (***Figure 8—figure supplement 3***).

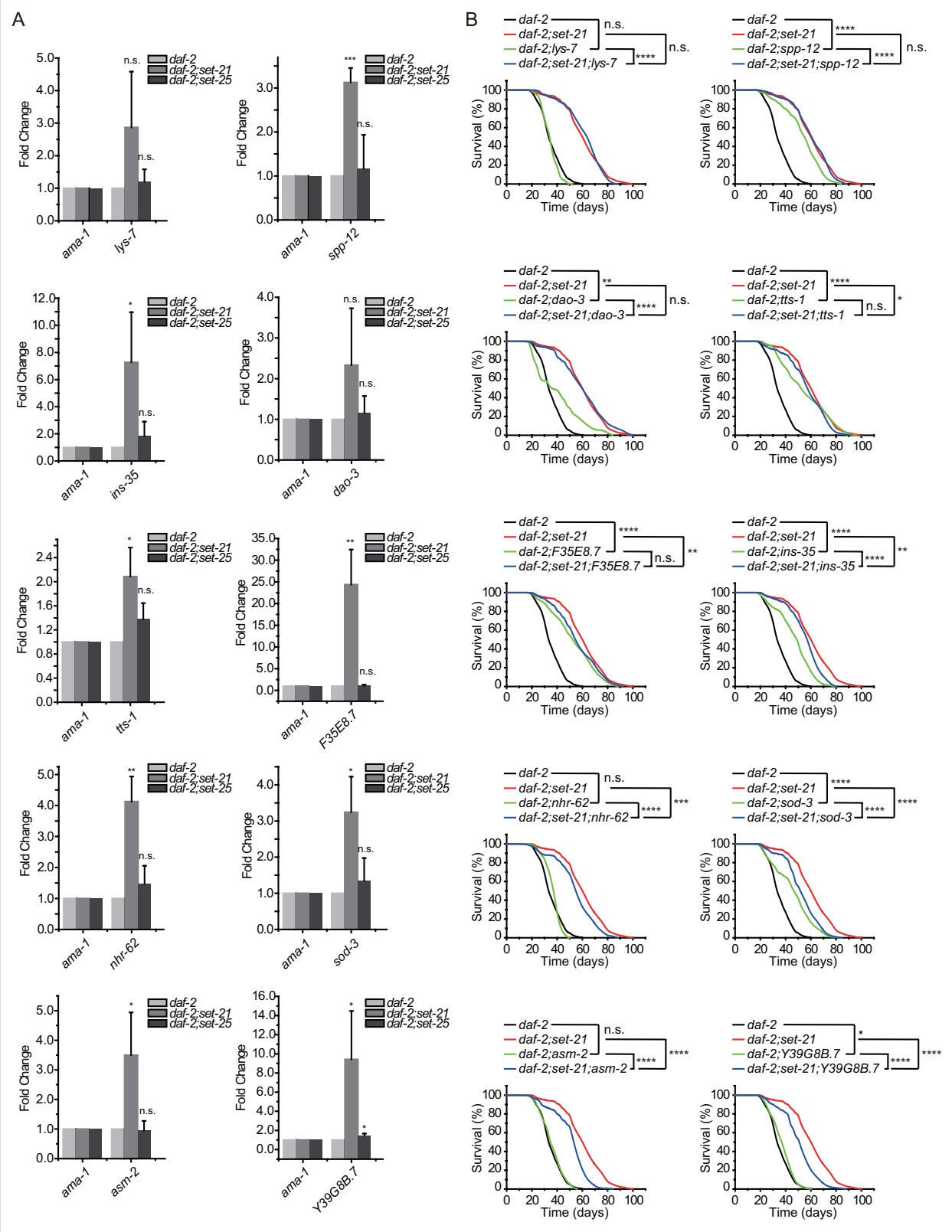

**Figure 7.** Class I DAF-16 genes are required for the synergistic lifespan extension in *daf-2;set-21* animals. (**A**) Quantitative real-time PCR analysis of the indicated mRNAs. Levels of *ama-1* mRNA were used as an internal control for sample normalization. Data were expressed as fold changes relative to those of *daf-2 (e1370)* animals. Data are presented as the mean ± s.e.m. of three independent experiments. Asterisks indicate significant differences using two-tailed t tests. *p<0.05; **p<0.01; ***p<0.001; ****p<0.0001; n.s., not significant, p>0.05. (**B**) Survival curves of indicated animals. Asterisks

*Figure 7 continued on next page*

Figure 7 continued

indicate significant differences using log rank tests. *p<0.05; **p<0.01; ***p<0.001; ****p<0.0001; n.s., not significant, p>0.05. Lifespan data were summarized in *Supplementary file 1*.

The online version of this article includes the following figure supplement(s) for figure 7:

**Figure supplement 1.** Gene structure and the alleles of indicated genes.

HMT G9a, also known as euchromatic histone lysine methyltransferase 2 (EHMT2), is the human homolog of SET-6 (*Yuan et al., 2020*) and mediates H3K9 dimethylation. Treating *daf-2* animals with the G9a inhibitor A-366 reduced H3K9me2 levels (*Figure 8B*, *Figure 8—figure supplement 4A-B*). The A-366 treatment extended the lifespan of *daf-2* worms by 15% (*Figure 8C*), but did not significantly change the lifespan of N2 worms (*Figure 8—figure supplement 4C*). Moreover, A-366 also increased nematode resistance to oxidative and heat stress of *daf-2* animals (*Figure 8D–E*). Thus, we concluded that decreased H3K9me1/2 levels in *daf-2* mutant by a G9a inhibitor may increase lifespan and resistance to oxidative stress and heat stress in *C. elegans*.

## Discussion

Here, we identified a class of putative HMTs, as well as methyltransferases that have been suggested to indirectly regulate H3K9me, that limit the lifespan of *daf-2* mutant in *C. elegans*. These putative HMTs are required for H3K9me1/2 modification and regulate DAF-16 Class I genes in *daf-2* animals. In the absence of DAF-2 and H3K9me1/2, the DAF-16 accumulates in the nucleus and the expression of DAF-16 Class I genes increase, which promote the expression of longevity genes and anti-stress genes and extends lifespan of *daf-2* animals (*Figure 8—figure supplement 4D*). Decreased H3K9me1/2 in *daf-2* animals improved nematode tolerance to oxidative stress and heat stress. Notably, the inhibition of worm H3K9me1/2 methyltransferases by a human G9a inhibitor also extends the lifespan of *daf-2* animals. Thus, targeting insulin pathway and H3K9me1/2 modification together may be a new means for the treatment of aging and aging-related diseases.

The depletion of SET proteins or the inhibition of worm H3K9me1/2 methyltransferases by G9a inhibitor significantly extended the lifespan only in *daf-2* mutant worms, but has modest effects on N2 animals. Consistently, the long-lived *set-6, set-19, set-20, set-21, set-32,* and *set-33* mutants only significantly decreased global H3K9me1/2 levels in *daf-2* mutants but not in N2 animals. These results together suggested that DAF-2 limits the effects of these SET proteins.

It is unclear how and why these SET proteins only limit lifespan in *daf-2* animals. There are three possibilities. First, DAF-2 may have profound effect on the chromatin accessibility. In *daf-2* mutants, the chromatin may be more compact and prohibit the activity of the SET proteins to conduct H3K9me2 modification. Second, *daf-2* mutation may sensitize the animals by increasing the nuclear accumulated DAF-16. Although it is unclear whether DAF-16 preferentially recognizes H3K9me2 or H3K9me3, we observed a much stronger enrichment of DAF-16 in the nucleus in *daf-2;set-21* than in *daf-2* animals, which suggested that H3K9me2 may prohibit the accumulation of DAF-16 in the nucleus. Third, DAF-2 may have other indirect effect on the SET proteins. Further ChIP experiments can be used to test whether *daf-2* mutation could affect the association of the SET proteins with chromatin. The ATAC-seq technology may allow us to investigate whether *daf-2* mutation have changed chromosome accessibility to facilitate the activity of SET proteins. In addition, identification of the substrate of SET proteins may help to elucidate their mechanism and functions.

In *daf-2* animals, DAF-16 modestly accumulated in the nucleus, as shown by the newly generated single-copy GFP-tagged DAF-16 transgene. Yet in *daf-2;set-21* double mutants, DAF-16 was strongly enriched in the nucleus. The depletion of DAF-16 completely blocked the synergistic lifespan extension in *daf-2;set-21* animals. The mRNA-seq experiments revealed that the mRNA levels of DAF-16 Class I genes were elevated in the long-lived *daf-2;set-19*, *daf-2;set-21*, and *daf-2;set-32* double mutants compared to those in control *daf-2* and short-lived *daf-2;set-25* mutants. Together, these data suggested that the activity to DAF-16 to Class I genes not only depends on the absence of insulin pathway, but also depends on the depletion of these SET proteins and H3K9me2. The single-copy GFP-tagged DAF-16 transgene may allow us to re-visit the mechanism of lifespan regulation pathways.

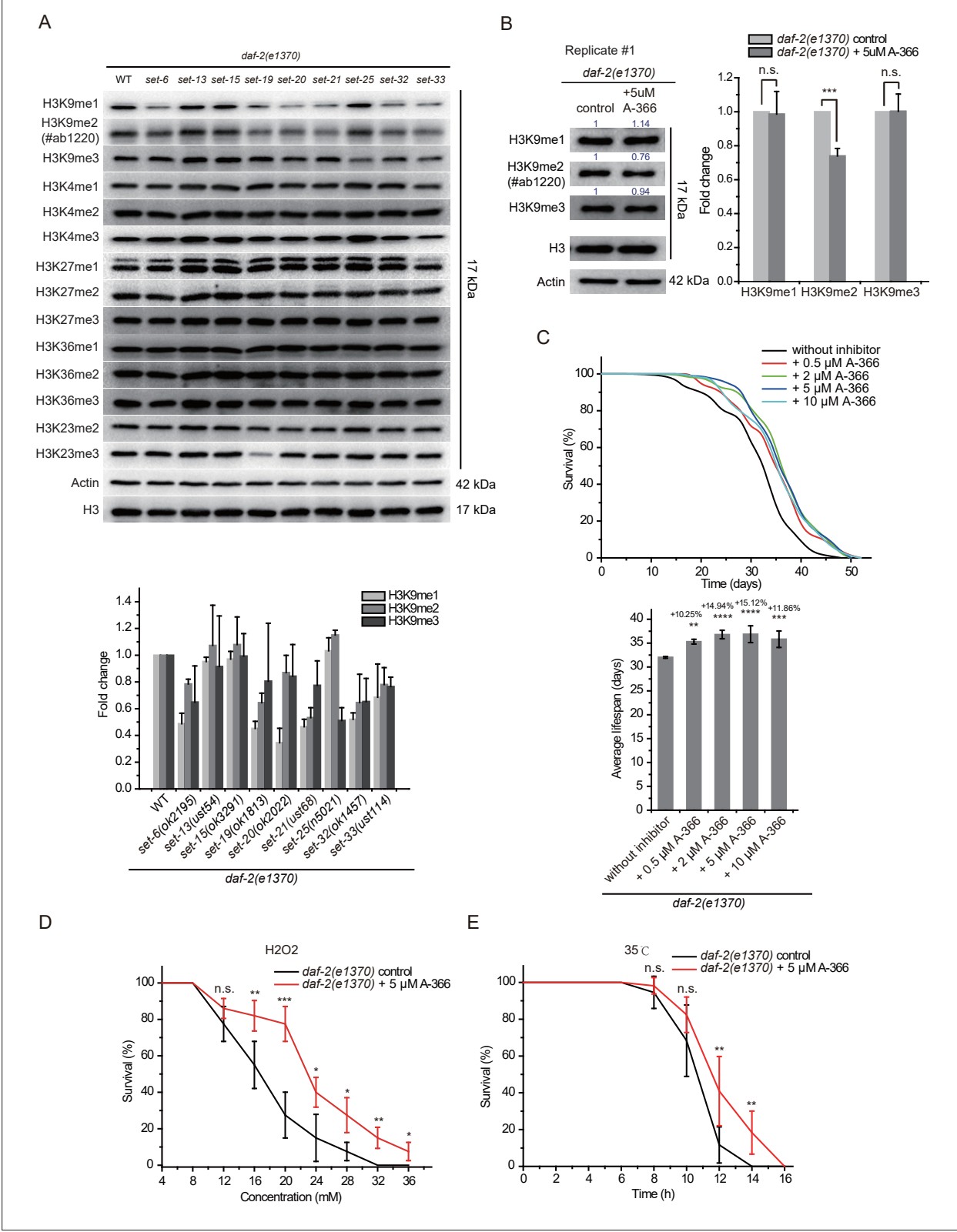

**Figure 8.** H3K9me1/2 methylation limits the lifespan and stress resistance of *daf-2* mutants. (**A**) (Top) Western blotting of L4 stage animals with the indicated antibodies (also see *Figure 8—figure supplement 1*). (Bottom) The histogram displayed means + s.e.m. of scanned density by ImageJ from three independent experiments. (**B**) The G9a (EHMT2) inhibitor A-366 reduced H3K9me2 levels in *daf-2* animals. (Left) Western blotting of L4 stage animals with the indicated antibodies. (Right) The histogram displayed means + s.e.m. of scanned density by ImageJ from three independent

*Figure 8 continued on next page*

*Figure 8 continued*

experiments. Data are expressed as fold changes relative to those of *daf-2(e1370)* animals without A-366. Asterisks indicate significant differences using two-tailed t tests. \*\*\*p<0.001. Also see *Figure 8—figure supplement 4A-B* for replicates. (**C**) (Top) Survival curves of indicated animals. (Bottom) Histogram displaying the average lifespan of the indicated animals. mean ± s.e.m. of three independent experiments. The percentage of change was compared to the average lifespan of the *daf-2* mutant. Asterisks indicate significant differences using log rank tests. 0.001 < \*\*p<0.01; \*\*\*p<0.001; \*\*\*\*p<0.0001. Lifespan data were summarized in *Supplementary file 1*. (**D, E**) Survival curves of G9a(EHMT2) inhibitor A-366-treated *daf-2* animals upon (**D**) oxidative and (**E**) heat stress. Data are presented as the mean ± s.e.m. of five independent experiments (n=50 animals). Asterisks indicate significant differences using two-tailed t tests. \*p<0.05; \*\*p<0.01; \*\*\*p<0.001; \*\*\*\*p<0.0001; n.s., not significant, p>0.05.

The online version of this article includes the following source data and figure supplement(s) for figure 8:

**Source data 1.** Original files and figures of the blots in *Figure 8A and B*.

**Figure supplement 1.** Western blotting of (**A and B**) L4 and (**C**) embryos with indicated antibodies.

**Figure supplement 2.** Western blotting of (**A**) L4 and (**B**) embryos with indicated antibodies.

**Figure supplement 3.** Chromatin immunoprecipitation (ChIP) of histone methylation marks in the indicated genes at the L4 stage.

**Figure supplement 4.** A-366 treatment reduced H3K9me2 levels in *daf-2* animals.

A trade-off has been proposed between fertility and lifespan, where increasing lifespan usually comes at the cost of reduced fertility (*Kirkwood, 1977*). In *C. elegans*, removing the germ cells extends lifespan, triggers the nuclear localization of the DAF-16/FOXO transcription factor, and activates target genes in the intestine (*Berman and Kenyon, 2006*). Some long-lived mutant worms, like *daf-2* and *rsks-1*, show reduced brood size (*Chen et al., 2013*). Consistently, we found that the long-lived worms daf-2;met-2, daf-2;set-6, daf-2;set-19, daf-2;set-20, daf-2;set-21, daf-2;set-32, and daf-2;set-33 all have reduced brood size compared to that of *daf-2* animals. However, additional investigations are necessary to examine whether removing the germline cells could further extend worm lifespan beyond the 100 days' ceiling.

Lifespan is controlled by both genetic and epigenetic factors. The perturbation of insulin/insulin-like signaling, TOR pathway, and mitochondrial functions have been shown to extensively modulate the aging process and lifespan. Epigenetic marks, including histone acetylation and methylation, are altered during aging and regulate lifespan in a number of species (*Lee et al., 2000*; *Lund et al., 2002*; *Bennett-Baker et al., 2003*; *Lu et al., 2004*). In *C. elegans*, both H3K4 and H3K27 methylation are involved in lifespan limitation. Animals with reduced H3K4me, for example, in *wdr-5*, *ash-2*, and *set-2* mutants, live longer than wild-type animals (*Greer et al., 2010*). Interestingly, perturbation of H3K27 methylation, via either a decrease in *mes-2* mutants or an increase in *utx-1* mutants, is associated with enhanced longevity, suggesting context-dependent lifespan regulation (*Maures et al., 2011*; *Guillermo et al., 2021*).

H3K9me3 is a hallmark of heterochromatin and is often enriched on repetitive sequences (*Peters et al., 2003*; *Guenatri et al., 2004*; *Martens et al., 2005*). In addition, H3K9me1/2/3 have all been shown enrichment on repeat sequences (*Gerstein et al., 2010*). H3K9 methylation and its association with the nuclear envelope help maintain silencing of repetitive sequences and may contribute to the maintenance and stability of heterochromatin-like regions in *C. elegans* (*Liu et al., 2011*), as observed in other organisms (*Peng and Karpen, 2008*). A hallmark of heterochromatin is the enrichment of HP1 (*James and Elgin, 1986*). The nematode *C. elegans* has two HP1 paralogous proteins: HPL-1 and HPL-2 (*Couteau et al., 2002*). Interestingly, HPL-2 may serve additional roles than HPL-1, as *hpl-2* mutants display diverse defects while *hpl-1* mutants generally lack obvious defects (*Garrigues et al., 2015*). The pattern of HPL-2 most closely resembles H3K9me1 and H3K9me2 but less closely resembles H3K9me3 (*Garrigues et al., 2015*). HPL-2-bound genes, especially those located on arms, possess significantly higher levels of H3K9me2, but show no significant difference in H3K9me3 levels when compared to genes not bound by HPL-2 (*Garrigues et al., 2015*). SET-25 is a putative H3K9me3 methyltransferase and is recruited to targeted genomic loci either by H3K9me2 which is deposited by MET-2 or by the somatic Argonaute protein NRDE-3 and small RNAs (*Mao et al., 2015*; *Padeken et al., 2021*). SET-25 colocalizes with both H3K9me3 and HPL-1, but does not colocalize with HPL-2 (*Towbin et al., 2012*).

H3K9me2 and H3K9me3 exhibit different patterns of repeat enrichment (*McMurchy et al., 2017*). H3K9me2 is more associated with DNA transposons and satellite repeats, similar to the heterochromatin factors HPL-2, LET-418, LIN-13, LIN-61, and MET-2, whereas H3K9me3 is particularly associated

with retrotransposon families, especially LINE and SINE elements (*McMurchy et al., 2017*). Moreover, H3K9me2 and the five heterochromatin factors are enriched at telomeres, whereas H3K9me3 is not (*McMurchy et al., 2017*). Therefore, H3K9me2, rather than H3K9me3, may be more closely associated with canonical heterochromatin factors in *C. elegans*. It was postulated that losing repressive chromatin is detrimental to lifespan (*Villeponteau, 1997*; *Tsurumi and Li, 2012*). In humans, two premature aging diseases are caused by mutations in lamins that reduce heterochromatin and disrupt its nuclear localization (*Shumaker et al., 2006*). In *C. elegans*, *Drosophila*, and mammals, heterochromatin decreases as individuals grow older (*Haithcock et al., 2005*; *Wood et al., 2010*; *Wood and Helfand, 2013*). Additionally, mutations that increase repressive chromatin extend lifespan in eukaryotes (*Kennedy et al., 1995*; *Jin et al., 2011*; *Maures et al., 2011*; *Larson et al., 2012*). In N2 background animals, the depletion of *hpl-2* was shown increased lifespan (*Meister et al., 2011*). However, we did not observe a further lifespan extension in *daf-2;hpl-2* animals, comparing to the *daf-2* mutant. Similarly, we did not observe a significant change in lifespan in *daf-2;set-25* mutants. The depletion of the two heterochromatin factors HPL-1 and HPL-2 failed to further extend the lifespan in *daf-2* mutant worms, suggesting that heterochromatin itself may not be sufficient to perturb lifespan in *daf-2* mutant worms.

MET-2, a mammalian H3K9 methyltransferase SETDB1 homolog (*Loyola et al., 2006*), likely monomethylates and dimethylates H3K9 in *C. elegans* (*Towbin et al., 2012*). MET-2 is necessary both for a normal lifespan in N2 background (*Tian et al., 2016*) and for the lifespan extension of *wdr-5* mutants (*Lee et al., 2019*). In *met-2* mutants, the lifespan is shortened compared to that of N2 animals. However, the lifespan was significantly extended in *daf-2;met-2* double mutants, suggesting context-dependent lifespan regulation by epigenetic machinery. SET-6 is a putative H3K9me2/3 methyltransferase that was postulated to prevent healthy aging rather than modulate the normal lifespan in N2 background (*Yuan et al., 2020*). SET-6 accelerates behavioral deterioration in *C. elegans* by reducing mitochondrial function and repressing the expression of nuclear-encoded mitochondrial proteins. Here, we found that the depletion of *set-6* dramatically increased the lifespan of *daf-2* animals, which further supports the context-dependent lifespan regulation model.

We noticed that the stress resistance is enhanced in both *met-2* and *daf-2;met-2* animals, yet *met-2* single mutant shortens lifespan and *daf-2;met-2* extends lifespan. In addition, *daf-2;set-15* showed a modest increase of stress resistance, while *daf-2;set-15* revealed no further lifespan extension. The reason is unknown yet. Previous research showed correlations between stress resistance and lifespan extension in many mutant worms, for example *daf-2*. However, since *daf-2* mutation already twitched the homeostasis and increased the stress resistance of animals, it is unclear that to what extent this correlation can further increase lifespan and stress resistance. Alternatively, some of these putative lifespan-limiting HMTs may have additional functions. Further investigations are required to understand the mechanism of each putative lifespan-limiting HMT.

In this work, we examined the role of 10 putative H3K9 methyltransferases, most of which have not been directly characterized for their biochemical properties by in vitro experiments (*Schwartz-Orbach et al., 2020*). The presence of these many putative H3K9 methyltransferases, or methyltransferases that may indirectly regulate H3K9me, may lead to a complexity of the interaction network of histone modifications. In addition, some SET proteins may have multiple roles in mediating histone modification. For example, SET-19 is required for both H3K9me1/2 and H3K23me3 marks, further bringing up complexity of the regulation. Further investigations are necessary to reveal their enzymatic activities by in vitro experiments (*Schwartz-Orbach et al., 2020*). In addition, identifying the in vivo targeted genes will also be crucial to help understanding why and how these SET proteins are involved in lifespan regulation in *daf-2* mutants.

Additionally, we identified seven genes required for H3K9me1/2-involved lifespan extension in *daf-2;set-21* animals. Among them, *tts-1* is a long noncoding RNA on ribosomes of the *daf-2* mutant. Depleting *tts-1* in *daf-2* mutants increases ribosome levels (*Tepper et al., 2013*; *Essers et al., 2015*). NHR-62 is known to be required for longevity in *C. elegans* (*Heestand et al., 2013*). INS-35 is an insulin-like peptides and may act on DAF-2 to regulate UPR$^{mt}$ and mitochondrial dynamics (*Chen et al., 2021*). SOD-3 is a superoxide dismutase that is involved in the removal of superoxide radicals. ASM-2 is an ortholog of human acid sphingomyelinase (ASM) and is involved in ceramide biosynthetic processes and sphingomyelin catabolic processes (*Lin et al., 1998*; *Deng et al., 2008*). The *C. elegans* genome encodes three ASM homologs, *asm-1*, *asm-2*, and *asm-3*. Among them, ASM-3

is most closely related to human ASM. During development and aging, ceramide and sphingosine accumulate (*Sacket et al., 2009*). Interestingly, monounsaturated fatty acid (MUFA) accumulation is necessary for the lifespan extension of H3K4me3-methyltransferase-deficient worms, and dietary MUFAs are sufficient to extend lifespan (*Han et al., 2017*). However, when *asm-1, asm-2, or asm-3* was inactivated by RNAi knockdown, a modest lifespan extension phenotype was observed (*Kim and Sun, 2012*). Further investigation is required to examine how and why these genes are involved in lifespan regulation in *daf-2;set-21* animals.

Stichodactyla toxin (ShK, ShkT) is a 35-residue basic peptide from the sea anemone *Stichodactyla helianthus* that blocks a number of potassium channels with nanomolar to picomolar potency. We found that F35E8.7 and Y39G8B.7 are required for lifespan extension in *daf-2;set-21* animals. Human proteins containing ShK-like domains are MMP-23 (matrix metalloprotease 23) and MFAP-2 (microfibril-associated glycoprotein 2). Whether modulating the expression of these two proteins or their downstream targets can improve longevity or healthy lifespan will stimulate much drug potential.

## Materials and methods

### Strains

Bristol strain N2 was used as the standard wild-type strain. All strains were grown at 20°C unless specified. The strains used in this study are listed in *Supplementary file 2*.

### Lifespan assay

Lifespan assays were performed at 20°C. Worm populations were synchronized by placing young adult worms on NGM plates seeded with the *Escherichia coli* strain OP50-1 (unless otherwise noted) for 4–6 hr and then removed. The hatching day was counted as day 1 for all lifespan measurements. Worms were transferred every other day to new plates to eliminate confounding progeny. Animals were scored as alive or dead every 2 (before 52 days) or 4 (after 52 days) days. Worms were scored as dead if they did not respond to repeated prods with a platinum pick. Worms were censored if they crawled off the plate or died from vulval bursting and bagging. Plates were censored if they were contaminated by fungi or mites. For each lifespan assay, 90 worms were used in 3 plates (30 worms/plate). All lifespan data were summarized in *Supplementary file 1*.

### Brood size

L4 hermaphrodites were singled onto plates and transferred daily as adults until embryo production ceased and the progeny numbers were scored.

### Hydrogen peroxide assay

Ten synchronized worms at day 1 of adulthood were transferred to each well, which contained 1 ml of worm S-basal buffer with various concentrations of $H_2O_2$ in a 12-well plate at 20°C. Four hours later, 100 µl of 1 mg/ml catalase (Sigma, C9322) was added to neutralize $H_2O_2$, and the mortality of worms was scored.

### Heat-shock assay

Ten synchronized hermaphrodites at day 1 of adulthood were transferred to new NGM plates and incubated at 35°C, and their mortality was checked every 2 or 4 hr until there were no living worm.

### G9a inhibitor treatment

Several P0 young adult worms were placed onto fresh NGM plates seeded with *E. coli* strain OP50 (with the same concentration of G9a inhibitors in both NGM plates and OP50 liquid). Four or five days later, F1 young adult worms were placed onto new NGM plates with OP50 and G9a inhibitor. Then, we singled out the F2 worms for other assays. For heat-shock assay, we placed worms on G9a plates until day 1 of adulthood, and then transferred the hermaphrodites to NGM plates without G9a inhibitor, and incubated the worms at 35°C.

### Western blot

Embryos or L4 stage worms were harvested and washed three times with M9 buffer. Samples were frozen in –80°C. Ten minutes at 95°C in ×1 protein dye (62.5 mM Tris pH 6.8, 10% glycerol, 2% SDS,

5% β-mercaptoethanol, 0.2% bromophenol blue) was sufficient to expose worm proteins. The next step was spin for 1 min at high speed to remove insoluble components, then quickly transfer supernatant into a new tube (on ice) and immediately run on gel or store aliquots at –80°C. Proteins were resolved by SDS-PAGE on gradient gels (10% separation gel, 5% spacer gel) and transferred to a Hybond-ECL membrane. After washing with ×1 TBST buffer (Sangon Biotech, Shanghai) and blocking with 5% milk-TBST, the membrane was incubated overnight at 4°C with antibodies (listed below). The membrane was washed three times for 10 min each with ×1 TBST and then incubated with secondary antibodies at room temperature for 2 hr. The membrane was washed three times for 10 min with ×1 TBST and then visualized.

The primary antibodies used were β-actin (Beyotime, AF5003), H3 (Abcam, ab1791), H3K4me1 (Abcam, ab176877), H3K4me2 (Abcam, ab32356), H3K4me3 (Abcam, ab8580), H3K9me1 (Abcam, ab9045), H3K9me2 #1 (Abcam, ab1220), H3K9me2 #2 (Abcam, ab176882), H3K9me3 (Millipore, 07-523), H3K27me1 (Abcam, ab194688), H3K27me2 (Abcam, ab24684), H3K27me3 (Millipore, 07-449), H3K36me1 (Abcam, ab9048), H3K36me2 (Abcam, ab9049), H3K36me3 (Abcam, ab9050), H3K23me2 (Active Motif, 39653), and H3K23me3 (Active Motif, 61499). The secondary antibodies used were goat anti-mouse (Beyotime, A0216) and goat anti-rabbit (Abcam, ab205718) antibodies.

## Construction of deletion mutants

For gene deletions, triple sgRNA-guided chromosome deletion was conducted as previously described (*Chen et al., 2014*). To construct sgRNA expression vectors, the 20 bp *unc-119* sgRNA guide sequence in the *pU6::unc-119* sgRNA(F+E) vector was replaced with different sgRNA guide sequences. Addgene plasmid #47549 was used to express Cas9 II protein. Plasmid mixtures containing 30 ng/μl of each of the three or four sgRNA expression vectors, 50 ng/μl Cas9 II-expressing plasmid, and 5 ng/μl pCFJ90 were coinjected into *tofu-5::gfp::3xflag (ustIS026)* animals. Deletion mutants were screened by PCR amplification and confirmed by sequencing. The sgRNA sequences are listed in *Supplementary file 3*.

## Construction of plasmids and transgenic strains

For SET-13::GFP, a SET-13 promoter and CDS region were PCR-amplified with the primers 5′-CTTA TAATACGACTCACTAGTTTGGGAAATTCACCGAAAAGGTAAC-3′ and 5′-CTCCACCTCCACCTCC GCGTACATTCTCCTTACCATC-3′ from N2 genomic DNA. A GFP::3xFLAG region was PCR-amplified with the primers 5′-GGAGGTGGAGGTGGAGCTATG-3′ and 5′-CTTGTCATCGTCATCCTTGTAATCG-3′ from plasmid pSG085. A SET-13 3′ UTR (untranslated region) was PCR-amplified with the primers 5′-GGATGACGATGACAAGTAATAAGATTGATAATTGATAATTGATAATGATTAAC-3′ and 5′-AGGAATTC CTCGAGACGTACGCCTGAAAATCTCCAAAATGTCAG-3′ from N2 genomic DNA. The ClonExpress MultiS one-step cloning kit (Vazyme, C113-02) was used to insert the SET-13::GFP::3x FLAG fusion gene into the pCFJ151 vector. The transgene was integrated onto the *C. elegans* chromosome IV by the MosSCI method (*Frøkjaer-Jensen et al., 2008*).

For the in situ transgene expressing 3x FLAG::GFP::SET-21, a 1.5 kb left arm was PCR-amplified with primers 5′-GGGTAACGCCAGCACGTGTGTGACATCGACGAGAGCAACA-3′ and 5′-GGCATCTA CAGGCATTGGAG-3′ from N2 genomic DNA. A 1.5 kb right arm was PCR-amplified with primers 5′-GGAGGTGGAGGTGGAGCTATGCCTGTAGATGCCGAGA-3′ and 5′-CAGCGGATAACAATTTCACA CTTCTTGTGCGGCAATAACA-3′ from N2 genomic DNA. A 3xflag::gfp insertion was PCR-amplified with the primers 5′-ATGGACTACAAAGACCATGACG-3′ and 5′-ATAGCTCCACCTCCACCTC-3′ from YY178 genomic DNA. The sgRNA sequences were listed in *Supplementary file 3*. ClonExpress MultiS One Step Cloning Kit (Vazyme C113-02, Nanjing) was used to connect these fragments with vector which is amplified with 5′-TGTGAAATTGTTATCCGCTGG-3′ and 5′-CACACGTGCTGGCGTTACC-3′ from L4440. The injection mix contained PDD162 (50 ng/μl), SET-21 repair plasmid (50 ng/μl), pCFJ90 (5 ng/μl), and three sgRNAs (30 ng/μl). The mix was injected into young adult N2 animals. The transgenes were integrated onto the *C. elegans'* chromosome IV by CRISPR/Cas9 system.

For the in situ transgene expressing 3x FLAG::GFP::SET-32, a 1.5 kb left arm was PCR-amplified with primers 5′-GGGTAACGCCAGCACGTGTCCATAATAGACGTCGTCG-3′ and 5′-CGATTTCG ATGACATCTTA-3′ from N2 genomic DNA. A 1.5 kb right arm was PCR-amplified with primers 5′-GAGGTGGAGGTGGAGCTATGTCATCGAAATCGAAGTC-3′ and 5′-CAGCGGATAACAATTTCACA TGAAAACGAGATGTACGTGA-3′ from N2 genomic DNA. A 3xflag::gfp insertion was PCR-amplified

with the primers 5'-ATGGACTACAAAGACCATGACG-3' and 5'-ATAGCTCCACCTCCACCTC-3' from YY178 genomic DNA. The sgRNA sequences were listed in *Supplementary file 3*. ClonExpress MultiS One Step Cloning Kit (Vazyme C113-02, Nanjing) was used to connect these fragments with vector which is amplified with 5'-TGTGAAATTGTTATCCGCTGG-3' and 5'-CACACGTGCTGG CGTTACC-3' from L4440. The injection mix contained PDD162 (50 ng/μl), SET-32 repair plasmid (50 ng/μl), pCFJ90 (5 ng/μl), and three sgRNAs (30 ng/μl). The mix was injected into young adult N2 animals. The transgenes were integrated onto the *C. elegans'* chromosome I by CRISPR/Cas9 system.

For the in situ transgene expressing SET-25::GFP::3x FLAG, a 1.5 kb left arm was PCR-amplified with primers 5'-gggtaacgccagCACGTGtgATTTTTCTCGCAAAGACGTG-3' and 5'-ATAGCTCCACCT CCACCTCCGAATGCAGGAAGGGTTCC-3' from N2 genomic DNA. A 1.5 kb right arm was PCR-amplified with primers 5'-ACAAGGATGACGATGACAAGTGAAAGTTTCTCCCCTATAC-3' and 5'-cagc ggataacaatttcacaCTTATCGTGTCGAGACCGG-3' from N2 genomic DNA. A GFP::3xFLAG region was PCR-amplified with the primers 5'-GGAGGTGGAGGTGGAGCTATGAGTAAAGG-3' and 5'-CTTG TCATCGTCATCCTTGTAATCG-3' from SHG326 genomic DNA. The sgRNA sequences were listed in *Supplementary file 3*. ClonExpress MultiS One Step Cloning Kit (Vazyme C113-02, Nanjing) was used to connect these fragments with vector which is amplified with 5'-TGTGAAATTGTTATCCGCTGG-3' and 5'-CACACGTGCTGGCGTTACC-3' from L4440. The injection mix contained PDD162 (50 ng/μl), SET-25 repair plasmid (50 ng/μl), pCFJ90 (5 ng/μl), and three sgRNAs (30 ng/μl). The mix was injected into young adult N2 animals. The transgenes were integrated onto the *C. elegans'* chromosome III by CRISPR/Cas9 system.

For the in situ transgene expressing DAF-16::GFP::3x FLAG, a 1.5 kb left arm was PCR-amplified with primers 5'-gggtaacgccagCACGTGtgTCTCGATTCCTGGATCGTCG-3' and 5'-ATAGCTCCACCT CCACCTCCCAAATCAAAATGAATATGCTGC- 3' from N2 genomic DNA. A 1.5 kb right arm was PCR-amplified with primers 5'-AGGATGACGATGACAAGTAAATTCTCTTCATTTTGTTTCCC-3' and 5'-cagc ggataacaatttcacaGCCAGAAAGAATTGGAAGGA-3' from N2 genomic DNA. A GFP::3xFLAG region was PCR-amplified with the primers 5'-GGAGGTGGAGGTGGAGCTATGAGTAAAGG-3' and 5'-CTTG TCATCGTCATCCTTGTAATCG-3' from SHG326 genomic DNA. The sgRNA sequences were listed in *Supplementary file 3*. ClonExpress MultiS One Step Cloning Kit (Vazyme C113-02, Nanjing) was used to connect these fragments with vector which is amplified with 5'-TGTGAAATTGTTATCCGCTGG-3' and 5'-CACACGTGCTGGCGTTACC-3' from L4440. The injection mix contained PDD162 (50 ng/μl), DAF-16 repair plasmid (50 ng/μl), pCFJ90 (5 ng/μl), and three sgRNAs (30 ng/μl). The mix was injected into young adult N2 animals. The transgenes were integrated onto the *C. elegans'* chromosome I by CRISPR/Cas9 system.

## RNA isolation

Synchronized L4 worms were sonicated in sonication buffer (20 mM Tris-HCl [pH 7.5], 200 mM NaCl, 2.5 mM MgCl$_2$, and 0.5% NP40). The eluates were incubated with TRIzol reagent followed by isopropanol precipitation and DNase I digestion. mRNA was purified from total RNA using poly-T oligo-attached magnetic beads. Sequencing was performed with a HiSeqTen instrument reading 150 base paired-end reads.

## RNA-seq analysis

The Illumina-generated raw reads were first filtered to remove adaptors, low-quality tags, and contaminants to obtain clean reads at Novogene. The clean reads were mapped to the reference genome of ce10 via TopHat software (version 2.1.1). Gene expression levels were determined by the fragments per kilobase of transcript per million mapped reads (FPKM).

## qRT-PCR for mRNA

Total RNA was reverse-transcribed into cDNA using the GoScript Reverse Transcription System (Promega) and quantified by qPCR using SYBR GREEN mix (Vazyme Q111-02, Nanjing) with a MyIQ2 real-time PCR system. Levels of *ama-1* mRNA were used as internal controls for sample normalization. Data are expressed as fold changes relative to those of *daf-2(e1370)* animals. The data analysis was performed using a ΔΔCT approach. The primers used for qRT-PCR are listed in *Supplementary file 4*.

## ChIP-qPCR

ChIP experiments were performed as previously described with L4 staged animals. After crosslinking, samples were resuspended in 1 ml FA buffer (50 mM Tris/HCl [pH 7.5], 1 mM EDTA, 1% Triton X-100, 0.1% sodium deoxycholate, and 150 mM NaCl) with a proteinase inhibitor tablet (Roche no. 05056489001) and sonicated for 20 cycles at high output (each cycle: 30 s on and 30 s off) with a Bioruptor plus. Lysates were precleared and then immunoprecipitated with 2 μl anti-histone H3 (monomethyl K9) antibody (Abcam no. ab9045), 2 μl anti-histone H3 (dimethyl K9) antibody (Abcam no. mAbcam1220) or 2 μl anti-trimethylated H3K9 antibody (Millipore no. 07-523). ChIP signals were normalized to coimmunoprecipitated *ama-1* and then expressed as fold changes relative to that of *daf-2 (e1370)* animals. qRT-PCR primers for ChIP assays are listed in *Supplementary file 5*.

## ChIP-seq data analysis

ChIP-seq reads were aligned to the WBcel235 assembly of the *C. elegans* genome using Bowtie2 version 2.3.5.1 (*Langmead and Salzberg, 2012*) by Ben Langmead with the default settings. The SAMtools version 0.1.19 (*Li et al., 2009*) 'view' utility was used to convert the alignments to BAM format, and the 'sort' utility was used to sort the alignment files. ChIP-seq peaks were called using MACs2 with default parameters against the ChIP-seq input sample. Custom Shell scripts were used to convert BAM files to BigWig format. Finally, deepTools version 3.4.3 (*Ramírez et al., 2014*) were used to generate profile plots showing the ChIP-seq signals of transcription factors around TSSs of indicated genes. The ChIP-seq datasets were downloaded from the ENCODE or NCBI GEO databases (*Supplementary file 6*).

## Statistics

Bar graphs with error bars are presented as the mean and standard deviation. All of the experiments were conducted with independent *C. elegans* animals for the indicated N times. Statistical analysis was performed with a two-tailed Student's t-test.

## Acknowledgements

We are grateful to the members of the Guang lab for their comments. We are grateful to the International *C. elegans* Gene Knockout Consortium, and the National Bioresource Project for providing the strains. Some strains were provided by the CGC, which is funded by NIH Office of Research Infrastructure Programs (P40 OD010440). This work was supported by grants from the Strategic Priority Research Program of the Chinese Academy of Sciences (XDB39010600), the National Key R&D Program of China (2019YFA0802600), the National Natural Science Foundation of China (91940303, 31870812, 32070619, 31871300, and 31900434). This study was supported by the Fundamental Research Funds for the Central Universities.

## Additional information

### Funding

| Funder | Grant reference number | Author |
|---|---|---|
| Fundamental Research Funds for the Central Universities | | Shouhong Guang |
| National Key Research and Development Program of China | 2019YFA0802600 | Shouhong Guang |
| National Natural Science Foundation of China | 91940303 | Shouhong Guang |
| National Natural Science Foundation of China | 31870812 | Shouhong Guang |

| Funder | Grant reference number | Author |
|---|---|---|
| National Natural Science Foundation of China | 32070619 | Xuezhu Feng |
| National Natural Science Foundation of China | 31871300 | Xuezhu Feng |
| National Natural Science Foundation of China | 31900434 | Chengming Zhu |

The funders had no role in study design, data collection and interpretation, or the decision to submit the work for publication.

## Author contributions

Meng Huang, Conceptualization, Formal analysis, Investigation, Visualization, Methodology, Writing - original draft, Project administration; Minjie Hong, Formal analysis, Validation, Investigation, Visualization, Methodology; Xinhao Hou, Software, Formal analysis, Visualization; Chengming Zhu, Data curation, Supervision; Di Chen, Resources, Methodology; Xiangyang Chen, Data curation, Supervision, Writing - review and editing; Shouhong Guang, Conceptualization, Resources, Supervision, Funding acquisition, Methodology, Writing - original draft, Project administration, Writing - review and editing; Xuezhu Feng, Supervision, Project administration, Writing - review and editing

## Author ORCIDs

Minjie Hong (iD) http://orcid.org/0000-0003-1510-6396
Di Chen (iD) http://orcid.org/0000-0002-0514-7947
Xiangyang Chen (iD) http://orcid.org/0000-0001-6530-7191
Shouhong Guang (iD) http://orcid.org/0000-0001-7700-9634

## Decision letter and Author response

Decision letter https://doi.org/10.7554/eLife.74812.sa1
Author response https://doi.org/10.7554/eLife.74812.sa2

---

# Additional files

## Supplementary files

• Supplementary file 1. Statistical analyses of lifespan experiments.

• Supplementary file 2. List of strains used in this study.

• Supplementary file 3. sgRNA sequences for CRISPR/Cas9-directed gene editing technology.

• Supplementary file 4. List of primers used in mRNA qPCR.

• Supplementary file 5. List of primers used in chromatin immunoprecipitation (ChIP)-qPCR.

• Supplementary file 6. Published chromatin immunoprecipitation sequencing (ChIP-seq) datasets used in the study.

• Transparent reporting form

• Source code 1. Code used to convert BAM files to BigWig format in ChIP-seq data analysis.

## Data availability

Sequencing data have been deposited in GSA under accession codes CRA005256. Figure 2 - Source Data 1 and Figure 8 - Source Data 2 contain the original files and figures of the blots used to generate the figures. Source Code Files 1 is used to convert BAM files to BigWig format in ChIP-seq data analysis.

The following dataset was generated:

| Author(s) | Year | Dataset title | Dataset URL | Database and Identifier |
|---|---|---|---|---|
| Guang S, Huang M, Hong M | 2021 | H3K9me1/2 methylation limits the lifespan of C. elegans | https://download.cncb.ac.cn/gsa/CRA005256 | GSA, CRA005256 |

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
