## [Editor Report]

This work presents important findings which pertain to how the length of an organism's lifespan can be affected by the presence of post-translational histone modifications, namely histone H3 lysine 9 mono- or di-methylation (H3K9me1/2). The authors find that the presence or absence of these marks plays a key role in modulating lifespan and stress resistance that synergizes with a long-lived insulin-pathway (daf-2) model. This work adds an important layer to aging regulation in organisms where a repressive chromatin state in specific genomic regions may limit the extent of lifespan. Hence, proper elucidation of this mechanism would be valuable to attain a comprehensive understanding of aging control and the reasons why these limits exist.

---

## [Decision Letter]

**Decision letter after peer review:**

Thank you for submitting your article "H3K9me1/2 methylation limits the lifespan of *C. elegans*" for consideration by *eLife*. Your article has been reviewed by 3 peer reviewers, and the evaluation has been overseen by a Reviewing Editor and Carlos Isales as the Senior Editor. The following individual involved in review of your submission has agreed to reveal their identity: David Katz (Reviewer #3).

Summary of reviewer discussion:

The reviewers were all very positive about the central idea of the manuscript, that putative HMTs act synergistically with daf-2 to extend lifespan. During review and subsequent discussion, several key weaknesses of the manuscript were identified that will need to be addressed before the manuscript is suitable for publication at *eLife*. While many of these weaknesses can be addressed by modifying the text and including additional supplementary information likely on hand, they also include a limited number of experiments that are necessary to support the key findings of the manuscript. Of highest importance, revisions must address the lack of wild-type and mutant only controls in methylation studies and the assumption of the enzymes studied as being bona fide H3K9 methyltransferases. A revision should focus on what is being shown (changes in H3K9me1/2 that correspond with lifespan in daf-2 mutants), while either addressing or taking into account the remaining unknowns. Additionally, all the reviewer comments are included, from which the essential revisions were identified, that should be addressed before publication.

Essential revisions:

1. The 6 genes being studied are putative methyltransferases, but are frequently described in the text as though they are validated and bona fide H3K9 methyltransferases. While the data presented support that they affect H3K9 methylation status, there are both direct and indirect ways this could occur. Some evidence that these putative HMTs are actual H3K9me1/me2 MTs (not acting indirectly, as observed in Schwartz-Orbach L et al., *eLife* 2020) and that they and daf-2 affect H3K9 methylation (ChIP seq) would greatly strengthen the manuscript. Barring this, the circumstantial data using inhibition would need to be strengthened and the text modified to clarify the lack of evidence (e.g., the introduction currently states that MET-2 and SET-25 are responsible for the *methylation* of H3K9, rather than saying the genes are required for the *presence* of H3K9me in vivo) on the direct nature of the observed effects.

2. To properly interpret the data, wild-type and mutant controls in methylation assays must be included. Presumably, based on the lifespan results, daf-2 changes methylation status in such a way that these genes are now limiting lifespan. Without control data, it is very difficult to interpret how this may work.

3. An explanation for why 6 different H3K9me1/me2 methylases would have such similar phenotypes. This explanation could come out of #1 and/or could involve other experiments and insights, including a better discussion of the individual HMTs and how they are similar/different. For example, addressing the H3K23me2/3 data in light of the Gu lab paper referenced in #1 may help in emphasizing the complexity of the interaction network and will also highlight the biological relevance of these findings.

4. Better discussion and clarity on the relationship between brood size and fertility tradeoffs in these animals, including inclusion of daf-2 single mutant brood sizes.

5. Better clarity on replication, blinding, and inclusion of summary survival data including conditions examined per assay, N, animals censored, median lifespan (along with average lifespan), and comparison used for determination of significance. If replicates are lacking, additional replicates may be necessary (see reviewer comments).

6. A significantly more detailed transcriptomic analysis and clarity about the analysis and it's methods: e.g., unbiased analysis to identify other pathways, differentiation between DAF-16 class I/II genes, better evidence to claim class I.

7. Tempered statements on generality, or more experiments to show generality, and clarified writing to more accurately report findings.

8. Use of the data to clarify the relationship between heterochromatin, H3K9me1/2, and H3K9me3.1.

*Reviewer #1 (Recommendations for the authors):*

1) The *C. elegans* enzymes described in the intro are still only putative methyltransferases. Most of them, outside of SET-2 have not been shown by in vitro methylation assays so should be referred to as putative methyltransferases (unless the authors wish to perform in vitro methylation assays to complement the in vivo findings of others which could still be indirect). Since so much of the manuscript is focused on H3K9 methylation it would be good to show that some of these enzymes (maybe just SET-21 if it is too much to show for them all) are indeed H3K9 methyltransferases.

2) It's important to see how H3K9 methylation changes in daf-2 worms relative to WT worms, both by WB at gross level and genome wide by ChIPseq. This change in H3K9me1/me2 genome wide should then be compared to daf-16 target gene lists to see if there is an enrichment.

3) For 6b you have to present all 4 conditions for epistasis experiments. The reader needs to see what the effect of lys-7 deletion is in the daf-2 background by itself. And then 2-way anova needs to be done to test whether the effect of lys-7 in the daf-2 background is different from in the daf-2;set-21 background.

4) Some attempt should be made to explain why there are 6 redundant H3K9me1/me2 methylases (again I'm not convinced yet that there are because of point 1) that all affect lifespan in the same manner. Are these enzymes expressed in the same tissue? Do they affect H3K9me1/me2 on the same genes? At the same time? Usually it is believed that different enzymes that modify the same mark have some sort of specificity but here they are not seeing that. I don't think it's necessary to perform tissue specific experiments for this initial manuscript but some experimental justification should be provided. Do they believe that it's the amount of heterochromatin DNA in general which is regulating this? Or is it a gene specific effect?

*Reviewer #2 (Recommendations for the authors):*

1. As the authors mentioned in the discussion, H3K9me2 is the histone modification in *C. elegans* that corresponds best with canonical heterochromatin factors like HPL-2 (McMurchy et al., 2017). However, in their results, they focus on the lack of effect on lifespan seen for H3K9me3 modifiers like SET-25 to rule out any effect with heterochromatin. In fact, despite corresponding highly with H3K9me2, one of the marks implicated in further extending the lifespan of their HMT; daf-2 double mutants, hpl-1 and hpl-2 mutations had no effect in a daf-2 background (although they do seem to slightly decrease lifespan, and it would be interesting to see the statistical comparison of a log-rank test). The authors should clarify the relationship between heterochromatin, H3K9me1/2, and H3K9me3.

2. I was confused by the statements describing Figure 4C on lines 176-178. Both double and triple mutants were much longer-lived than daf-2 single mutants, rather than having similar lifespan extensions as the daf-2 single mutants. There was even a distinction between the daf-2; set-21 double mutant and the daf-2; set-21; set-2 triple mutant – the presence of a set-2 mutation represses the slight lifespan extension difference seen in the daf-2; set-21 double mutant when compared to the daf-2; set-2 double mutant. The authors should rephrase their description of their results to more accurately convey their findings.

3. My major concerns with the approach to transcriptomic analysis were conveyed in the public review, but I include more specific concerns here. Given that Class I genes are an important downstream consequence of DAF-16, it isn't clear why these are primarily unaffected in the daf-2 control condition shown in Figure 5A. Furthermore, the authors focus primarily on the top 50 Class I genes, but there are interesting effects seen in the top 50 Class II genes as well, especially given that mutations in the HMTs seem to suppress some of the upregulation observed in daf-2 control – why exclude these from their interpretations? Additionally, the brevity of the methods makes it hard to tell exactly what comparisons are being drawn – in the heatmaps shown in in Figure 5C, D, does the color refer upreglation in comparison to a wild-type control? Why not compare misregulation to a daf-2 single mutant instead? What does "Top 50" refer to – are these the fifty most-affected in these specific data sets, or are they a gene set identified by an independent metric? Why limit the analysis to only fifty genes? Finally, this analysis would be further strengthened by comparing locus-specific effects to the known locations of H3K9me1/2/3 peaks in wild-type genomes in previously published datasets (like in Li et al., 2021 or McMurchy et al., 2017) or even to H3K9me1/2/3 localization in datasets of relevant HMT mutants like set-32.

4. How were the 10 Class I DAF-16 target genes chosen? It doesn't seem like they were in the top 50 (or not that I could easily see). Furthermore, why were no Class II genes chosen? The authors should clarify how they chose to examine these specific candidates, and if necessary, extend the analysis for other targets to see whether the effects are specific to Class I targets only (as they argue in their model). Furthermore, each of these candidate genes needs to be examined in a daf-2 mutant background to fully interpret the meaning of the suppression of long-lived daf-2; set-21 mutants.

5. The authors claim that loss of H3K9me1/2 increases binding of DAF-16 at its target genes (line 282). There is no evidence to support this claim (and a lack of evidence for this claim, if we take the DAF-16 localization data in Figure S3 at complete face value). This could be part of their proposed model (outlined in Figure S7), but as written, it reads as if it were a summary of work presented in the paper. The authors should clarify their writing to ensure full accuracy for what their data can support.

6. Throughout the manuscript, it's difficult to build a cohesive understanding of the functional effects of the HMT mutants. In part, this is because the paper investigates several HMTs without discussing their specific effects (in certain tissues or cell types, or at specific stages). The analyses are sometimes too broad (immunoblots of total histone modifications) and at other times too narrow (focusing only on 50 Class I DAF-16 target genes). Given that H3K9me1/2 will likely have locus specificity, it is unsatisfying to leave the analysis at the global level of immunoblots (although it is very compelling to see global effects in their mutants). Including a deeper, and unbiased, analysis of their transcriptomic data will help to bridge the gap between the global and the locus-specific, and may even point out other interesting avenues that will further bolster their claims about potential mechanism (or open new avenues of study).

[Editors’ note: further revisions were suggested prior to acceptance, as described below.]

Thank you for resubmitting your work entitled "H3K9me1/2 methylation limits the lifespan of *C. elegans*" for further consideration by *eLife*. Your revised article has been evaluated by Carlos Isales (Senior Editor) and a Reviewing Editor.

The manuscript has been improved but there are some remaining issues that need to be addressed, as outlined below:

Primarily, the three areas that remain an issue before submission are (1) a second replicate needs to be included (and ideally, performed blinded) for experiments such as Figures 1B, 4A, and 4B, (2) the appropriate epistasis controls should be included as requested by reviewer 1 to properly interpret the experiments (e.g., Figure 7), and (3) the text needs to be modified where precision remains an issue, as described by reviewer 1. There should be no technical issues with completing these experiments, and they're relatively straightforward, although I appreciate that it can take some time to complete longevity assays for these long-lived mutants. However, these experiments are necessary to experimentally validate the claims within the manuscript, along with responses to the reviewer comments below.

*Reviewer #1 (Recommendations for the authors):*

In the revised manuscript "H3K9me1/2 methylation limits the lifespan of *C. elegans*" Huang M, Hong M et al., performed additional experiments to examine the role of putative H3K9 methyltransferases in the lifespan regulation of daf-2 mutant worms. While the new experiments do bolster some portions of the manuscript I felt that the critical questions which I had asked in my initial review weren't addressed and are still important. I understand that COVID has made it difficult to perform certain experiments but with those restrictions in mind then the paper needs to be reworked to reflect what the experiments actually show rather than the model the authors wish to be true. Most importantly the role of H3K9 methylation in daf-2 mutants is still not really clear. These genes seem to have virtually no effect in WT background so the text needs to really be rewritten to reflect that. Starting with the title of the manuscript and then from the intro on it should be written from the perspective of "we wanted to examine how chromatin modifying enzymes were important in insulin receptor mutant worms".

1) As stated last review (major point #3), in epistasis experiments (such as WT and eat-2 +/- set-21 and WT and daf-2 +/- met-2, and Figure 3C +D, 7B) you need to include all 4 conditions in the experiment to make any conclusions. For example, you can't do the set-21 vs WT at a different time and assume that the set-21 is still going to have no effect in that particular experiment. There might be some unknown factor (food, temperature, etc) that makes set-21 actually have an effect in the WT worms in that experiment which would mean a very different conclusion. Then after the experiment is done you perform a 2 way ANOVA analysis to reveal whether set-21 has a significantly different effect in the WT context or the mutant context. This comment still holds true for asm-2, Y39G8B.7, etc… that I had asked for last review. You do not need to perform all possible combinations of these 3 genes but at the very least daf-2, daf-2;set-21, daf-2;asm-2 and daf-2;set-21;asm-2 if you want to make any conclusion about asm-2 relative to set-21 in the daf-2 background), examining 3 lines in an epistasis experiment rather than 4 is not really informative.

2) The text needs to be rewritten to reflect that this is all only occurring in daf-2 mutant worms and not in WT worms. This rewriting needs to start with the title ("…limits the lifespan of daf-2 mutant *C. elegans*".) and work its way thru the intro, results and discussion (where the authors claim H3K9me regulators should be considered for lifespan regulation). Need to include explicit writing that deletion of all these putative enzymes has only modest effects on H3K9me and lifespan in the WT context and their role is only revealed in the daf-2 mutants. Need to speculate about what that is. What makes the daf-2 mutation special to reveal a regulatory role for these genes?

*Reviewer #2 (Recommendations for the authors):*

Upon resubmission, this paper is greatly improved – readers now have the evidence needed to evaluate the paper's main claim that H3K9 HMTs synergistically extend lifespan in daf-2 mutants, in part via DAF-16, but also via. The additional experiments, along with a more tempered approach to discussing their findings, significantly strengthen the paper.

However, several of our concerns were not addressed by these changes, nor in the rebuttal. For example, Reviewer 2 asked that some of the key lifespan experiments be replicated at least once and also performed blinded – there are several figures where only a single replicate is shown. Key controls are still missing, like in Figure 7, where Reviewer 1 had asked for appropriate controls to interpret the epistasis experiments for example, set-21 single mutants and daf-2; lys-7 double mutants. The authors' rebuttal mentioned generating new alleles for some genes, but providing appropriate single mutant and double mutant controls should require no new strains to be built (since the strains likely already exist if the authors generated the triple mutant).

The extended analysis of the mRNA-seq is better (and opens up an interesting line of questioning for future studies!). However, the authors failed to address most of Reviewer 2's Major Point 3 about their transcriptomic analysis, nor did they include further details about how the analysis was done. The methods for the mRNA-seq will need to be more detailed. What is their unbiased approach? What is the differential expression shown in the Figure 5? For example, I had previously asked: "Does the color refer to upregulation in comparison to a wild-type control? Why not compare misregulation to a daf-2 single mutant instead?" and the answer is still not clear. Furthermore, if Class I genes are defined as being induced in daf-2 mutants, why are they downregulated in Figure 5E? And if Class II genes are defined as being not induced in daf-2 mutants, why are they upregulated in Figure 5F? I may be misunderstanding the gene classifications, but it is a demonstration of how the text is unclear. Overall, I found the new transcriptomic analyses are compelling, but more care needs to be taken to fully explain what was done.

---

## [Author Response]

Essential revisions:1. The 6 genes being studied are putative methyltransferases, but are frequently described in the text as though they are validated and bona fide H3K9 methyltransferases. While the data presented support that they affect H3K9 methylation status, there are both direct and indirect ways this could occur. Some evidence that these putative HMTs are actual H3K9me1/me2 MTs (not acting indirectly, as observed in Schwartz-Orbach L et al., eLife 2020) and that they and daf-2 affect H3K9 methylation (ChIP seq) would greatly strengthen the manuscript. Barring this, the circumstantial data using inhibition would need to be strengthened and the text modified to clarify the lack of evidence (e.g., the introduction currently states that MET-2 and SET-25 are responsible for the methylation of H3K9, rather than saying the genes are required for the presence of H3K9me in vivo) on the direct nature of the observed effects.

Thanks very much for the comments.

We completely agree with the reviewers that we lack direct biochemical evidence showing that these SET proteins are bona fide H3K9 methyltransferases. We have tried very hard to conduct similar experiments as Schwartz-Orbach has done on SET-32. However, given the covid-19 pandemic and biosafety issue, we could not get access to [3H]-labeled S-adenosylmethionine (SAM). We are very sorry that we could not pursue further at this stage. Therefore, we have revised the manuscript saying that these SET proteins are putative histone methyltransferases and are required for the presence of H3K9me1/2 marks. The text of MET-2 and SET-25 has been revised similarly and saying that they are required for the presence of H3K9me in vivo.

We also added in the Discussion section that further investigations are required to reveal their enzymatic activities by in vitro experiments (lines 496-506).

2. To properly interpret the data, wild-type and mutant controls in methylation assays must be included. Presumably, based on the lifespan results, daf-2 changes methylation status in such a way that these genes are now limiting lifespan. Without control data, it is very difficult to interpret how this may work.

Thanks very much for the comments.

We have included new data comparing the global levels of H3K9me1/2/3 in N2 and *daf-2* animals by Western blotting (new Figure 8—figure supplement 1A). The *daf2* mutation itself did not significantly change the gross levels of H3K9me1/2/3 status.

We also compared the levels of H3K9me1/2/3, H3K4me1/2/3, H3K27me1/2/3, H3K36me1/2/3 and H3K23me2/3 in N2 and *set* mutants at L4 and embryonic stages (new Figure 8—figure supplement 1B-C). In N2 background, the mutation of these *set* genes did not significantly change gross levels of H3K9me1/2/3 either.

We speculated that the mutation of daf-2 sensitizes the animals, for example,

increases the cytoplasm-to-nuclear translocation of DAF-16. We have generated a

single-copy, in situ GFP-tagged DAF-16 transgene, crossed to daf-2, set-21, set-25

single mutants and their double mutants, and scored the nuclear localization of DAF16 at 20oC and 25oC, respectively (new Figure 5A-B, Figure 5—figure supplement 1B). The mutation of daf-2 increased the nuclear accumulation of DAF-16. The double mutants daf-2;set-21 and daf-2;set-25 further increased the nuclear accumulation of DAF-16.

Interestingly, DAF-16 was nearly 100% localized in the nucleus in both *daf-2;set21* and *daf-2;set-25* double mutants. Yet *daf-2;set-21*, but not *daf-2;set-25*, extended the lifespan of *daf-2* animals, suggesting addition regulation required beyond the nuclear localization of DAF-16. Our work suggested that the transcription activity of DAF-16 may preferentially depend on H3K9me1/2, but not H3K9me3.

We have revised the text to include the information (lines 245-275, 357-361).

3. An explanation for why 6 different H3K9me1/me2 methylases would have such similar phenotypes. This explanation could come out of #1 and/or could involve other experiments and insights, including a better discussion of the individual HMTs and how they are similar/different. For example, addressing the H3K23me2/3 data in light of the Gu lab paper referenced in #1 may help in emphasizing the complexity of the interaction network and will also highlight the biological relevance of these findings.

Thanks very much for the comments. We are also eager to know why the 6 different H3K9me1/me2 methylases would have such similar phenotypes in this lifespan extension assay, yet have no clue at this stage.

We have generated in situ single-copy 3xFLAG-GFP tagged SET-21, SET-25 and SET-32 transgenes by CRISPR/Cas9 technology and single-copy GFP-3xFlAG-tagged SET-13 transgenic strains using the Mos1-mediated single-copy insertion (MosSCI) technology. These animals were imaged in N2 and *daf-2* background at embryonic and larva stages respectively, to figure out the expression patterns and cellular localizations of SET proteins (new Figure 4—figure supplement 1). However, we failed to find a clear correlation between their expression pattern with the lifespan extension phenotype. The depletion of SET-21 and SET-32 extended the lifespan of *daf-2* animals, but GFP::SET21 and GFP::SET-32 showed different expression patterns and subcellular localizations. GFP::SET-21 exclusively located in the nucleus but GFP::SET-32 located in both the nucleus and cytoplasm in embryos. At larval stage, GFP::SET-21 was expressed in soma but GFP::SET-32 was expressed in germline. Both *set-13* and *set-25* mutants did not show lifespan extension in N2 and *daf-2* background worms. SET-13 and SET-25 were enriched in the nucleus. SET-13::GFP was expressed through 8-cell stage to lateembryonic stage, while SET-25::GFP was expressed in the whole embryonic stage and the germline in young adults.

To identify the target genes in this group of lifespan extension *daf-2;set* mutants, we re-analyzed the mRNA expression profile in the double mutants via an unbiased method (Figure 6, Figure 6—figure supplement 1). We have identified 49 co-upregulated genes and 11 co-downregulated genes that are specifically enriched in the long-lived double mutants *daf-2;set-19*, *daf-2,set-21*, *daf-2;set-32*, but not in *daf-2* and *daf-2;daf25* animals (Figure 6A-B and Figure 6—figure supplement 1A). Interestingly, among the 49 co-upregulated genes, 27 of them are also DAF-16 Class I genes (Figure 6—figure supplement 1B). 22 co-upregulated genes are not DAF-16 Class I genes, suggesting the existence of additional regulations.

*C. elegans’* genome encodes 10 putative H3K9 methyltransferases, most of which have not been directly characterized for their biochemical properties by in vitro experiments. The presence of these many putative H3K9 methyltransferases may lead to a complexity of the interaction network of histone modifications. In addition, some SET proteins may have multiple roles in mediating histone modifications. For example, SET-19 is required for both H3K9me1/2 and H3K23me3 marks, further bringing up complexity of the regulation. Further investigations are necessary to reveal their enzymatic activities by in vitro experiments. In addition, identifying the in vivo targeted genes will also be crucial to help understanding why and how these SET proteins are involved in lifespan regulation.

We have revised the text to include the information (lines 186-201, 301-320, 496506).

4. Better discussion and clarity on the relationship between brood size and fertility tradeoffs in these animals, including inclusion of daf-2 single mutant brood sizes.

Thanks very much for the suggestion.

We have included the brood size assay of N2, *daf-2, set-21* and *daf-2;set-21*. Knocking out *set-21* reduced the brood size of *daf-2(e1370)* mutant worms and significantly extended lifespan (Figure 1B, Figure1—figure supplement 1C). However, we did not find significant difference between the brood size of N2 and *daf-2* (new Figure 1—figure supplement 1C and Figure 3—figure supplement 1B).

A trade-off has been proposed between fertility and lifespan, where increasing lifespan usually comes at the cost of reduced fertility. In *C. elegans*, removing the germ cells extends lifespan by triggering the nuclear localization of the DAF-16/FOXO transcription factor and activating target genes in the intestine. Some long-lived mutant worms, like *daf-2* and *rsks^-1^*, show reduced brood size. Consistently, we found that the long-lived worms daf-2;met-2, daf-2;set-6, daf-2;set-19, daf-2;set-20, daf-2;set-21, daf-2;set-32, and daf-2;set-33 all have reduced brood size compared to that of *daf-2* animals. However, additional investigations are necessary to examine whether removing the germline cells could further extend worm lifespan.

We revised the Discussion section to include this information (lines 115-117, 402411).

5. Better clarity on replication, blinding, and inclusion of summary survival data including conditions examined per assay, N, animals censored, median lifespan (along with average lifespan), and comparison used for determination of significance. If replicates are lacking, additional replicates may be necessary (see reviewer comments).

Thanks very much for the comments.

We have revised the Materials and methods section (lines 549-559) and figure legends to include the information of each life span assay. We also included a new Table S1 to summarize all lifespan experiments.

6. A significantly more detailed transcriptomic analysis and clarity about the analysis and it's methods: e.g., unbiased analysis to identify other pathways, differentiation between DAF-16 class I/II genes, better evidence to claim class I.

Thanks very much for the suggestions.

We used two approaches to analyze the mRNA-seq data.

First, the depletion of DAF-2 reduces the insulin signaling pathway, promotes DAF-16 nuclear translocation, and leads to both upregulation and downregulation of large sets of genes, referred to as Class I and II genes, respectively. Class I genes are induced in *daf-2* mutants but are repressed in *daf-2;daf-16* double mutants. Class II genes are not induced in *daf-2* mutants but are induced in *daf-2; daf-16* double mutants. 1663 genes are classified as positive (class I) DAF-16 targets and 1733 genes are classified as negative (class II) DAF-16 targets of DAF-16. Class I genes are enriched for the Gene Ontology categories including oxidation, reduction, and energy metabolism, whereas class II genes are enriched for genes involved in biosynthesis, growth, reproduction, and development. We performed mRNA-seq and analyzed Class I and II DAF-16 genes to identify the mis-regulated genes in the *daf-2;set* mutants. Interestingly, the mRNA levels of DAF-16 Class I, but not Class II, genes are consistently activated in long-lived *daf-2;set-19, daf-2;set-21* and *daf-2;set-32* worms, than in the control *daf-2* and *daf-2;set-25* animals (Figure 5E-F and new Figure 5figure supplement 2).

Second, to identify the target genes in the group of lifespan extension *daf-2;set* mutants, we re-analyzed the mRNA expression profile in the double mutants via an unbiased method (new Figure 6, Figure 6—figure supplement 1). We have identified 49 co-upregulated genes and 11 co-downregulated genes that are specifically enriched in the long-lived double mutants *daf-2;set-19*, *daf-2,set-21*, *daf-2;set-32*, but not in *daf-2* and *daf-2;daf-25* animals (Figure 6A-B and Figure 6—figure supplement 1A). Interestingly, among the 49 co-upregulated genes, 27 of them are also DAF-16 Class I genes (new Figure 6—figure supplement 1B). 22 co-upregulated genes are not DAF-16 Class I genes, suggesting the existence of additional regulations.

We then analyzed a number of known transcription factors for their binding to the co-regulated targeted genes (new Figure 6C-D, Figure 6—figure supplement 1C-E). Among them, we found that DAF-16 and NHR-80 were specifically enriched at the transcription start sites (TSS) of the 49 co-upregulated genes. NHR-80 is a homolog of mammalian hepatocyte nuclear factor 4 and is an important nuclear hormone receptor involved in the control of fat consumption and fatty acid composition in *C. elegans.* Among the genes targeted by DAF-16 and NHR-80, twelve of them are co-regulated by both factors, which is consistent with previous report that *daf-16* and *nhr-80* function in parallel pathway for lipid metabolism.

We revised the text to include this information (lines 284-320).

7. Tempered statements on generality, or more experiments to show generality, and clarified writing to more accurately report findings.

Thanks very much for the suggestion. We have revised the manuscript to tone down these SET proteins as putative H3K9me1/2 methyltransferases. And we also carefully revised the text to more accurately report our findings.

8. Use of the data to clarify the relationship between heterochromatin, H3K9me1/2, and H3K9me3.1.

Thanks very much for the suggestion. Actually, the relationship between heterochromatin, H3K9me1/2, and H3K9me3 is quite intriguing.

H3K9me3 is a hallmark of heterochromatin and is often enriched on repetitive sequences. In addition, H3K9me1/2/3 have all been shown enrichment on repeat sequences (GERSTEIN *et al.,* 2010). H3K9 methylation and its association with the nuclear envelope help maintain silencing of repetitive sequences and may contribute to the maintenance and stability of heterochromatin-like regions in *C. elegans*, as observed in other organisms. A hallmark of heterochromatin is the enrichment of HP1. The nematode *C. elegans* has two HP1 paralogous proteins: HP1 Like 1 and 2 (HPL-1 and HPL-2). Interestingly, HPL-2 may serve additional roles than HPL-1, as *hpl-2* mutants display diverse defects while *hpl-1* mutants generally lack obvious defects. The pattern of HPL-2 most closely resembles H3K9me1 and H3K9me2 but less closely resembles H3K9me3. HPL-2-bound genes, especially those located on arms, possess significantly higher levels of H3K9me2, but show no significant difference in H3K9me3 levels when compared to genes not bound by HPL-2. SET-25 is a putative H3K9me3 methyltransferase and is recruited to targeted genomic loci either by H3K9me2 which is deposited by MET-2 or by the somatic Argonaute protein NRDE-3 and small RNAs. SET-25 colocalizes with both H3K9me3 and HPL-1, but does not colocalize with HPL-

2.

H3K9me2 and H3K9me3 exhibit different patterns of repeat enrichment. H3K9me2 is more associated with DNA transposons and satellite repeats, similar to the heterochromatin factors HPL-2, LET-418, LIN-13, LIN-61, and MET-2, whereas H3K9me3 is particularly associated with retrotransposon families, especially LINE and SINE elements. Moreover, H3K9me2 and the five heterochromatin factors are enriched at telomeres, whereas H3K9me3 is not. Therefore, H3K9me2, rather than H3K9me3, may be more closely associated with canonical heterochromatin factors in *C. elegans*. It was postulated that losing repressive chromatin is detrimental to lifespan. In humans, two premature aging diseases are caused by mutations in lamins that reduce heterochromatin and disrupt its nuclear localization. In *C. elegans*, *Drosophila,* and mammals, heterochromatin decreases as individuals grow older. Additionally, mutations that increase repressive chromatin extend lifespan in eukaryotes. In N2 background animals, the depletion of *hpl-2* was shown increased lifespan. However, we did not observe a further lifespan extension in *daf-2;hpl-2* animals, comparing to the *daf-2* mutant. Similarly, we did not observe a significant change in lifespan in *daf2;set-25* mutants. The depletion of the two heterochromatin factors HPL-1 and HPL-2 failed to further extend the lifespan, suggesting that heterochromatin itself may not be sufficient to perturb lifespan in *daf-2* mutant worms.

We have revised the Discussion section to include the above information (lines 426-469).

Reviewer #1 (Recommendations for the authors):1) The *C. elegans* enzymes described in the intro are still only putative methyltransferases. Most of them, outside of SET-2 have not been shown by in vitro methylation assays so should be referred to as putative methyltransferases (unless the authors wish to perform in vitro methylation assays to complement the in vivo findings of others which could still be indirect). Since so much of the manuscript is focused on H3K9 methylation it would be good to show that some of these enzymes (maybe just SET-21 if it is too much to show for them all) are indeed H3K9 methyltransferases.

Thanks very much for the suggestion.

We completely agree with the reviewer that we lack direct evidence showing that these SET proteins are bona fide H3K9 methyltransferases. We have tried very hard to conduct similar experiments as Schwartz-Orbach et al., has done on SET-32. However, given the covid-19 pandemic and biosafety issue, we could not get access to [3H]labeled S-adenosylmethionine (SAM). We are very sorry that we could not pursue further at this stage. Therefore, we have revised the manuscript saying that these SET proteins are putative histone methyltransferases and are required for the presence of H3K9me1/2.

*C. elegans’* genome encodes 10 putative H3K9 methyltransferases, most of which have not been directly characterized for their biochemical properties by in vitro experiments (SCHWARTZ-ORBACH *et al.,* 2020). The presence of these many putative H3K9 methyltransferases may lead to a complexity of the interaction network of histone modifications. In addition, some SET proteins may have multiple roles in mediating histone modification. For example, SET-19 is required for both H3K9me1/2 and H3K23me3 marks, further bringing up complexity of the regulation. Further investigations are necessary to reveal their enzymatic activities by in vitro experiments

(SCHWARTZ-ORBACH *et al.,* 2020). In addition, identifying the in vivo targeted genes will also be crucial to help understanding why and how these SET proteins are involved in lifespan regulation.

We have revised the Discussion section to include the above information (lines 496-506).

2) It's important to see how H3K9 methylation changes in daf-2 worms relative to WT worms, both by WB at gross level and genome wide by ChIPseq. This change in H3K9me1/me2 genome wide should then be compared to daf-16 target gene lists to see if there is an enrichment.

Thanks very much for the comments.

We have included new data comparing the global levels of H3K9me1/2/3 in N2 and *daf-2* animals by Western blotting (new Figure 8—figure supplement 1A). The *daf2* mutation itself did not significantly change the gross levels of H3K9me1/2/3 status.

We also compared the levels of H3K9me1/2/3, H3K4me1/2/3, H3K27me1/2/3, H3K36me1/2/3 and H3K23me2/3 in N2 and *set* mutants at L4 and embryonic stages (new Figure 8—figure supplement 1B-C). In N2 background, the mutation of these *set* genes did not significantly change gross levels of H3K9me1/2/3 either.

However, for technical reasons, we could not successfully conduct ChIP-seq experiments on *daf-2* and *daf-2;set* larva animals.

We speculated that the mutation of *daf-2* sensitizes the animals, for example,

increases the cytoplasm-to-nuclear translocation of DAF-16. We have generated a

single-copy, in situ GFP-tagged DAF-16 transgene, crossed to *daf-2, set-21, set-25*

single mutants and their double mutants, and scored the nuclear localization of DAF16 at 20^o^C and 25^o^C, respectively (new Figure 5A-B, Figure 5—figure supplement 1B). The mutation of *daf-2* increased the nuclear accumulation of DAF-16. The double mutants *daf-2;set-21* and *daf-2;set-25* further increased the nuclear accumulation of DAF-16.

Interestingly, DAF-16 was nearly 100% localized in the nucleus in both *daf-2;set21* and *daf-2;set-25* double mutants. Yet *daf-2;set-21*, but not *daf-2;set-25*, extended the lifespan of *daf-2* animals, suggesting addition regulation required beyond the nuclear localization of DAF-16. Our work suggested that the transcription activity of DAF-16 may preferentially depend on H3K9me1/2, but not H3K9me3.

We have revised the text to include the information (lines 245-275, 357-361).

3) For 6b you have to present all 4 conditions for epistasis experiments. The reader needs to see what the effect of lys-7 deletion is in the daf-2 background by itself. And then 2-way anova needs to be done to test whether the effect of lys-7 in the daf-2 background is different from in the daf-2;set-21 background.

Thanks very much for the suggestions. These are really valuable experiments to understand the mechanism of each putative target gene. However, for the time issue, we are still at the stage to generate multiple alleles and transgenes of *nhr-62, sod-3, asm-2, F35E8.7,* and *Y39G8B.7*. Hopefully, we can carefully investigate how and why each gene is involved in lifespan regulation in *daf-2;set* double mutant animals in the near future.

4) Some attempt should be made to explain why there are 6 redundant H3K9me1/me2 methylases (again I'm not convinced yet that there are because of point 1) that all affect lifespan in the same manner. Are these enzymes expressed in the same tissue? Do they affect H3K9me1/me2 on the same genes? At the same time? Usually it is believed that different enzymes that modify the same mark have some sort of specificity but here they are not seeing that. I don't think it's necessary to perform tissue specific experiments for this initial manuscript but some experimental justification should be provided. Do they believe that it's the amount of heterochromatin DNA in general which is regulating this? Or is it a gene specific effect?

Thanks very much for the comments. We are also eager to know why the 6 different H3K9me1/me2 methylases would have such similar phenotypes in this lifespan extension assay, yet have no clue at this stage.

We have generated in situ single-copy 3xFLAG-GFP tagged SET-21, SET-25 and SET-32 transgenes by CRISPR/Cas9 technology and single-copy GFP-3xFlag-tagged SET-13 transgenic strains using the Mos1-mediated single-copy insertion (MosSCI) technology. These animals were imaged in N2 and *daf-2* background at embryonic and larva stages respectively, to figure out the expression patterns and cellular localizations of SET proteins (new Figure 4—figure supplement 1). However, we failed to find a clear correlation between their expression pattern with the lifespan extension phenotype. The depletion of SET-21 and SET-32 extended the lifespan of *daf-2* animals, but GFP::SET21 and GFP::SET-32 showed different expression pattern and subcellular localization. GFP::SET-21 exclusively located in the nucleus but GFP::SET-32 located in both the nucleus and cytoplasm in embryos. At larval stage, GFP::SET-21 was expressed in soma but GFP::SET-32 was expressed in germline. Both *set-13* and *set-25* mutants did not show lifespan extension in N2 and *daf-2* background worms. SET-13 and SET-25 were enriched in the nucleus. SET-13::GFP was expressed through 8-cell stage to lateembryonic stage, while SET-25::GFP was expressed in the whole embryonic stage and the germline in young adults.

To identify the target genes in the group of lifespan extension *daf-2;set* mutants, we re-analyzed the mRNA expression profile in the double mutants via an unbiased method (new Figure 6, Figure 6—figure supplement 1). We have identified 49 coupregulated genes and 11 co-downregulated genes that are specifically enriched in the long-lived double mutants *daf-2;set-19*, *daf-2,set-21*, *daf-2;set-32*, but not in *daf-2* and *daf-2;daf-25* animals (new Figure 6A-B and Figure 6—figure supplement 1A). Interestingly, among the 49 co-upregulated genes, 27 of them are also DAF-16 Class I genes (new Figure 6—figure supplement 1B). 22 co-upregulated genes are not DAF-16 Class I genes, suggesting the existence of additional regulations.

*C. elegans’* genome encodes 10 putative H3K9 methyltransferases, most of which have not been directly characterized for their biochemical properties by in vitro experiments. The presence of these many putative H3K9 methyltransferases may lead to a complexity of the interaction network of histone modifications. In addition, some SET proteins may have multiple roles in mediating histone modifications. For example, SET-19 is required for both H3K9me1/2 and H3K23me3 marks, further bringing up complexity of the regulation. Further investigations are necessary to reveal their enzymatic activities by in vitro experiments. In addition, identifying the in vivo targeted genes will also be crucial to help understanding why and how these SET proteins are involved in lifespan regulation.

We have revised the text to include the information (lines 186-201, 301-320, 496-506).

Reviewer #2 (Recommendations for the authors):1. As the authors mentioned in the discussion, H3K9me2 is the histone modification in *C. elegans* that corresponds best with canonical heterochromatin factors like HPL-2 (McMurchy et al., 2017). However, in their results, they focus on the lack of effect on lifespan seen for H3K9me3 modifiers like SET-25 to rule out any effect with heterochromatin. In fact, despite corresponding highly with H3K9me2, one of the marks implicated in further extending the lifespan of their HMT; daf-2 double mutants, hpl-1 and hpl-2 mutations had no effect in a daf-2 background (although they do seem to slightly decrease lifespan, and it would be interesting to see the statistical comparison of a log-rank test). The authors should clarify the relationship between heterochromatin, H3K9me1/2, and H3K9me3.

Thanks very much for the suggestion. Actually, the relationship between heterochromatin, H3K9me1/2, and H3K9me3 is quite intriguing.

H3K9me3 is a hallmark of heterochromatin and is often enriched on repetitive sequences. In addition, H3K9me1/2/3 have all been shown enrichment on repeat sequences. H3K9 methylation and its association with the nuclear envelope help maintain silencing of repetitive sequences and may contribute to the maintenance and stability of heterochromatin-like regions in *C. elegans*, as observed in other organisms. A hallmark of heterochromatin is the enrichment of HP1. The nematode *C. elegans* has two HP1 paralogous proteins: HP1 Like 1 and 2 (HPL-1 and HPL-2). Interestingly, HPL-2 may serve additional roles than HPL-1, as *hpl-2* mutants display diverse defects while *hpl-1* mutants generally lack obvious defects. The pattern of HPL-2 most closely resembles H3K9me1 and H3K9me2 but less closely resembles H3K9me3. HPL-2bound genes, especially those located on arms, possess significantly higher levels of H3K9me2, but show no significant difference in H3K9me3 levels when compared to genes not bound by HPL-2. SET-25 is a putative H3K9me3 methyltransferase and is recruited to targeted genomic loci either by H3K9me2 which is deposited by MET-2 or by the somatic Argonaute protein NRDE-3 and small RNAs. SET-25 colocalizes with both H3K9me3 and HPL-1, but does not colocalize with HPL-2.

H3K9me2 and H3K9me3 exhibit different patterns of repeat enrichment. H3K9me2 is more associated with DNA transposons and satellite repeats, similar to the heterochromatin factors HPL-2, LET-418, LIN-13, LIN-61, and MET-2, whereas H3K9me3 is particularly associated with retrotransposon families, especially LINE and SINE elements. Moreover, H3K9me2 and the five heterochromatin factors are enriched at telomeres, whereas H3K9me3 is not. Therefore, H3K9me2, rather than H3K9me3, may be more closely associated with canonical heterochromatin factors in *C. elegans*. It was postulated that losing repressive chromatin is detrimental to lifespan. In humans, two premature aging diseases are caused by mutations in lamins that reduce heterochromatin and disrupt its nuclear localization. In *C. elegans*, *Drosophila,* and mammals, heterochromatin decreases as individuals grow older. Additionally, mutations that increase repressive chromatin extend lifespan in eukaryotes. In N2 background animals, the depletion of *hpl-2* was shown increased lifespan. However, we did not observe a further lifespan extension in *daf-2;hpl-2* animals, comparing to the *daf-2* mutant. Similarly, we did not observe a significant change in lifespan in *daf2;set-25* mutants. The depletion of the two heterochromatin factors HPL-1 and HPL-2 failed to further extend the lifespan, suggesting that heterochromatin itself may not be sufficient to perturb lifespan in *daf-2* mutant worms.

We have revised the Discussion section to include the above information (lines 426469).

We have revised the Materials and methods section (lines 549-559) and figure legends to include the information of each lifespan assay. We also included a new Table S1 to summarize all lifespan experiments.

2. I was confused by the statements describing Figure 4C on lines 176-178. Both double and triple mutants were much longer-lived than daf-2 single mutants, rather than having similar lifespan extensions as the daf-2 single mutants. There was even a distinction between the daf-2; set-21 double mutant and the daf-2; set-21; set-2 triple mutant – the presence of a set-2 mutation represses the slight lifespan extension difference seen in the daf-2; set-21 double mutant when compared to the daf-2; set-2 double mutant. The authors should rephrase their description of their results to more accurately convey their findings.

Sorry for the confusion.

We have revised the text as “the double mutants *daf-2;set-2* and *daf-2;set-21* and

the triple mutants *daf-2;set-2;set-21* exhibited similar lifespan extensions relative to that of *daf-2* animals (Figure 4C)” (lines 221-223).

3. My major concerns with the approach to transcriptomic analysis were conveyed in the public review, but I include more specific concerns here. Given that Class I genes are an important downstream consequence of DAF-16, it isn't clear why these are primarily unaffected in the daf-2 control condition shown in Figure 5A. Furthermore, the authors focus primarily on the top 50 Class I genes, but there are interesting effects seen in the top 50 Class II genes as well, especially given that mutations in the HMTs seem to suppress some of the upregulation observed in daf-2 control – why exclude these from their interpretations? Additionally, the brevity of the methods makes it hard to tell exactly what comparisons are being drawn – in the heatmaps shown in in Figure 5C, D, does the color refer upreglation in comparison to a wild-type control? Why not compare misregulation to a daf-2 single mutant instead? What does "Top 50" refer to – are these the fifty most-affected in these specific data sets, or are they a gene set identified by an independent metric? Why limit the analysis to only fifty genes? Finally, this analysis would be further strengthened by comparing locus-specific effects to the known locations of H3K9me1/2/3 peaks in wild-type genomes in previously published datasets (like in Li et al., 2021 or McMurchy et al., 2017) or even to H3K9me1/2/3 localization in datasets of relevant HMT mutants like set-32.

Thanks very much for the suggestions.

We used two approaches to analyze the mRNA-seq data.

First, the depletion of DAF-2 reduces the insulin signaling pathway, promotes DAF-16 nuclear translocation, and leads to both upregulation and downregulation of large sets of genes, referred to as Class I and II genes, respectively. Class I genes are induced in *daf-2* mutants but are repressed in *daf-2;daf-16* double mutants. Class II genes are not induced in *daf-2* mutants but are induced in *daf-2; daf-16* double mutants. 1663 genes are classified as positive (class I) DAF-16 targets and 1733 genes are classified as negative (class II) DAF-16 targets of DAF-16. Class I genes are enriched for the Gene Ontology categories including oxidation, reduction, and energy metabolism, whereas class II genes are enriched for genes involved in biosynthesis, growth, reproduction, and development. We performed mRNA-seq and analyzed Class I and II DAF-16 genes to identify the mis-regulated genes in the *daf-2;set* mutants. Interestingly, the mRNA levels of DAF-16 Class I, but not Class II, genes are consistently activated in long-lived *daf-2;set-19, daf-2;set-21* and *daf-2;set-32* worms, than in the control *daf-2* and *daf-2;set-25* animals (Figure 5E-F and new Figure 5—figure supplement 2).

Second, to identify the target genes in the group of lifespan extension *daf-2;set* mutants, we re-analyzed the mRNA expression profile in the double mutants via an unbiased method (new Figure 6, Figure 6—figure supplement 1). We have identified 49 co-upregulated genes and 11 co-downregulated genes that are specifically enriched in the long-lived double mutants *daf-2;set-19*, *daf-2,set-21*, *daf-2;set-32*, but not in *daf-2* and *daf-2;daf-25* animals (Figure 6A-B and Figure 6—figure supplement 1A). Interestingly, among the 49 co-upregulated genes, 27 of them are also DAF-16 Class I genes (new Figure 6—figure supplement 1B). 22 co-upregulated genes are not DAF-16 Class I genes, suggesting the existence of additional regulations.

We then analyzed a number of known transcription factors for their binding to the co-regulated targeted genes (new Figure 6C-D, Figure 6—figure supplement 1C-E). Among them, we found that DAF-16 and NHR-80 were specifically enriched at the transcription start sites (TSS) of the 49 co-upregulated genes. NHR-80 is a homolog of mammalian hepatocyte nuclear factor 4 and is an important nuclear hormone receptor involved in the control of fat consumption and fatty acid composition in *C. elegans.* Among the genes targeted by DAF-16 and NHR-80, twelve of them are co-regulated by both factors, which is consistent with previous report that *daf-16* and *nhr-80* function in parallel pathway for lipid metabolism.

We revised the text to include this information (lines 284-320).

4. How were the 10 Class I DAF-16 target genes chosen? It doesn't seem like they were in the top 50 (or not that I could easily see). Furthermore, why were no Class II genes chosen? The authors should clarify how they chose to examine these specific candidates, and if necessary, extend the analysis for other targets to see whether the effects are specific to Class I targets only (as they argue in their model). Furthermore, each of these candidate genes needs to be examined in a daf-2 mutant background to fully interpret the meaning of the suppression of long-lived daf-2; set-21 mutants.

Thanks very much for the suggestion.

To identify the target genes in these group of lifespan extension *daf-2;set* mutants, we analyzed the mRNA expression profile in the double mutants via an unbiased method (new Figure 6, Figure 6—figure supplement 1). We have identified 49 coupregulated genes and 11 co-downregulated genes that are specifically enriched in the long-lived double mutants *daf-2;set-19*, *daf-2,set-21*, *daf-2;set-32*, but not in *daf-2* and *daf-2;daf-25* animals (Figure 6A-B and Figure 6—figure supplement 1A). Interestingly, among the 49 co-upregulated genes, 27 of them are also DAF-16 Class I genes (new Figure 6—figure supplement 1B). 22 co-upregulated genes are not DAF-16 Class I genes, suggesting the existence of additional regulations.

To confirm the change in target gene expression, we chose 10 genes from the 49 co-upregulated genes. Among them, *nhr-62, dao-3* and *tts^-1^* are direct binding targets of both DAF-16 and NHR-80. *sod-3* is a DAF-16 direct binding target. *lys-7* and *asm2* are NHR-80 direct binding targets. All of the 10 genes have been shown to be upregulated in *daf-2* compared to *daf-2;daf-16* animals. We quantified the mRNA levels by quantitative real-time PCR (qRT–PCR) in *daf-2*, *daf-2;set-21* and *daf-2;set-25* animals (Figure 7A). Consistently, all the 10 genes were up-regulated in *daf-2;set-21* double mutants than in *daf-2* and *daf-2;set-25* animals (Figure 7A).

We are very interested in how and why each gene is involved in lifespan regulation in *daf-2;set* double mutant animals, we are trying to generate multiple alleles and transgenes for *nhr-62, sod-3, asm-2, F35E8.7,* and *Y39G8B.7*. However, for the time issue, we are still at the very early stage to generate more lines. Hopefully, we can answer this question in the near future.

5. The authors claim that loss of H3K9me1/2 increases binding of DAF-16 at its target genes (line 282). There is no evidence to support this claim (and a lack of evidence for this claim, if we take the DAF-16 localization data in Figure S3 at complete face value). This could be part of their proposed model (outlined in Figure S7), but as written, it reads as if it were a summary of work presented in the paper. The authors should clarify their writing to ensure full accuracy for what their data can support.

Thanks very much for the comments. We completely agree with the reviewer that we lack direct data showing that these *set* mutations increased the binding activity of DAF-16 to target genes by ChIP analysis.

We speculated that the mutation of *daf-2* sensitizes the animals, for example, increases the cytoplasm-to-nuclear translocation of DAF-16. We have generated a single-copy, in situ GFP-tagged DAF-16 transgene, crossed to *daf-2, set-21, set-25* single mutants and their double mutants, and scored the nuclear localization of DAF16 at 20^o^C and 25^o^C, respectively (new Figure 5A-B, Figure 5—figure supplement 1B). The mutation of *daf-2* increased the nuclear accumulation of DAF-16. The double mutants *daf-2;set-21* and *daf-2;set-25* further increased the nuclear accumulation of DAF-16.

Interestingly, DAF-16 was nearly 100% localized in the nucleus in both *daf-2;set21* and *daf-2;set-25* double mutants. Yet *daf-2;set-21*, but not *daf-2;set-25*, extended the lifespan of *daf-2* animals, suggesting addition regulation required beyond the nuclear localization of DAF-16. Our work suggested that the transcription activity of DAF-16 may preferentially depend on H3K9me1/2, but not H3K9me3.

This result is unlike a previous widely used multi-copied DAF-16::GFP transgene. In the multi-copied DAF-16::GFP transgene animals, DAF-16::GFP localized in the cytoplasm and a *daf-2* mutation completely re-localizes DAF-16::GFP to the nucleus (Figure 5—figure supplement 1A). This result suggests further investigations are required to understand the mechanism of DAF-2/DAF-16 axis in lifespan regulation.

We have revised the text to include the information (lines 357-361).

We also revised the working model in the Discussion section (lines 391-396) as:

“These putative histone methyltransferases are required for H3K9me1/2 modification and regulate DAF-16 Class I genes. In the absence of H3K9me1/2, the DAF-16 accumulates in the nucleus and the expression of DAF-16 Class I genes increase, which promote the expression of longevity genes and anti-stress genes and extends lifespan of *daf-2* animals (Figure 8—figure supplement 4D).”

6. Throughout the manuscript, it's difficult to build a cohesive understanding of the functional effects of the HMT mutants. In part, this is because the paper investigates several HMTs without discussing their specific effects (in certain tissues or cell types, or at specific stages). The analyses are sometimes too broad (immunoblots of total histone modifications) and at other times too narrow (focusing only on 50 Class I DAF-16 target genes). Given that H3K9me1/2 will likely have locus specificity, it is unsatisfying to leave the analysis at the global level of immunoblots (although it is very compelling to see global effects in their mutants). Including a deeper, and unbiased, analysis of their transcriptomic data will help to bridge the gap between the global and the locus-specific, and may even point out other interesting avenues that will further bolster their claims about potential mechanism (or open new avenues of study).

Thanks very much for the suggestions.

We have generated in situ single-copy 3xFLAG-GFP tagged SET-21, SET-25 and SET-32 transgenes by CRISPR/Cas9 technology and single-copy GFP-3xFLAG-tagged SET-13 transgenic strains using the Mos1-mediated single-copy insertion (MosSCI) technology. These animals were imaged in N2 and *daf-2* background at embryonic and larva stages respectively, to figure out the expression patterns and cellular localizations of SET proteins (new Figure 4—figure supplement 1). However, we failed to find a clear correlation between their expression pattern with the lifespan extension phenotype. The depletion of SET-21 and SET-32 extended the lifespan of *daf-2* animals, but GFP::SET21 and GFP::SET-32 showed different expression patterns and subcellular localizations. GFP::SET-21 exclusively located in the nucleus but GFP::SET-32 located in both the nucleus and cytoplasm in embryos. At larval stage, GFP::SET-21 was expressed in soma but GFP::SET-32 was expressed in germline. Both *set-13* and *set-25* mutants did not show lifespan extension in N2 and *daf-2* background worms. SET-13 and SET-25 were enriched in the nucleus. SET-13::GFP was expressed through 8-cell stage to lateembryonic stage, while SET-25::GFP was expressed in the whole embryonic stage and the germline in young adults.

To identify the target genes in these group of lifespan extension *daf-2;set* mutants, we have analyzed the mRNA expression profile in the double mutants via an unbiased method (new Figure 6, Figure 6—figure supplement 1). We have identified 49 coupregulated genes and 11 co-downregulated genes that are specifically enriched in the long-lived double mutants *daf-2;set-19*, *daf-2,set-21*, *daf-2;set-32*, but not in *daf-2* and *daf-2;daf-25* animals (new Figure 6A-B and Figure 6—figure supplement 1A). Interestingly, among the 49 co-upregulated genes, 27 of them are also DAF-16 Class I genes (new Figure 6—figure supplement 1B). 22 co-upregulated genes are not DAF-16

Class I genes, suggesting the existence of additional regulations.

*C. elegans’* genome encodes 10 putative H3K9 methyltransferases, most of which have not been directly characterized for their biochemical properties by in vitro experiments. The presence of these many putative H3K9 methyltransferases may lead to a complexity of the interaction network of histone modifications. In addition, some SET proteins may have multiple roles in mediating histone modification. For example, SET-19 is required for both H3K9me1/2 and H3K23me3 marks, further bringing up complexity of the regulation. Further investigations are necessary to reveal their enzymatic activities by in vitro experiments. In addition, identifying the in vivo targeted genes will also be crucial to help understanding why and how these SET proteins are involved in lifespan regulation.

We have revised the text to include the information (lines 186-201, 301-320, 496506).

[Editors’ note: what follows is the authors’ response to the second round of review.]

The manuscript has been improved but there are some remaining issues that need to be addressed, as outlined below:Primarily, the three areas that remain an issue before submission are (1) a second replicate needs to be included (and ideally, performed blinded) for experiments such as Figures 1B, 4A, and 4B,

Thanks very much for the suggestions.

We have included a blinded replicate to compare the lifespan of *daf-2* vs *daf-2;set-21* mutants (new Figure 1A). We have also conducted new experiments to compare N2 vs all *set-21* alleles (new Figure 1A). We did not found significant lifespan difference between N2 and all *set-21* alleles, but *daf-2;set-21* double mutants strongly extended the lifespan of *daf-2* worms.

We have included blinded replicates for experiments in Figures 4A and 4B. The conclusions were unchanged.

2) the appropriate epistasis controls should be included as requested by reviewer 1 to properly interpret the experiments (e.g., Figure 7),

Thanks very much for the comments. We have included new epistasis control experiments and data analysis for Figures 1A, 1B, 3-S3, and 7B. The conclusions were unchanged.

and (3) the text needs to be modified where precision remains an issue, as described by reviewer 1.

Thanks very much for the comments. We have revised the manuscript following reviewer #1’ s suggestions to state that the lifespan extension effect is likely more specific to *daf-2(-)* background animals.

Reviewer #1 (Recommendations for the authors):1) As stated last review (major point #3), in epistasis experiments (such as WT and eat-2 +/- set-21 and WT and daf-2 +/- met-2, and Figure 3C +D, 7B) you need to include all 4 conditions in the experiment to make any conclusions. For example, you can't do the set-21 vs WT at a different time and assume that the set-21 is still going to have no effect in that particular experiment. There might be some unknown factor (food, temperature, etc) that makes set-21 actually have an effect in the WT worms in that experiment which would mean a very different conclusion. Then after the experiment is done you perform a 2 way ANOVA analysis to reveal whether set-21 has a significantly different effect in the WT context or the mutant context. This comment still holds true for asm-2, Y39G8B.7, etc… that I had asked for last review. You do not need to perform all possible combinations of these 3 genes but at the very least daf-2, daf-2;set-21, daf-2;asm-2 and daf-2;set-21;asm-2 if you want to make any conclusion about asm-2 relative to set-21 in the daf-2 background), examining 3 lines in an epistasis experiment rather than 4 is not really informative.

Thanks very much for the suggestions.

­

We have conducted new experiments to compare N2 vs all *set-21* alleles (new Figure 1B). We did not found significant lifespan difference between N2 and all *set-21* alleles, but *daf-2;set-21* double mutants strongly extended the lifespan of *daf-2* worms.

In new Figure 1C, we also included N2 vs *set-21(ust68)*. We did not found significant lifespan difference between N2 and *set-21(ust68)* mutants, but *eat-2;set-21* double mutants strongly extended the lifespan of *eat-2* worms.

We conducted new experiments to compare the stress resistance of N2 vs *set* mutants (new Figure 3-S3A-B). We did not find strong difference of stress resistance between N2 vs *set* mutants, yet there are significant difference of stress resistance between *daf-2* vs *daf-2;set* mutants (new Figure 3D-E).

We included new replicates for Figure 4A-B. The conclusions are not changed.

We conducted new experiments to compare the lifespan of *daf-2*, *daf-2;y39g8b.7, daf-2;set-21* double mutants *vs daf-2;set-21;y39g8b.7* triple mutant and etc… (new Figure 7B). Although we did not find significant lifespan difference between *daf-2* vs *daf-2;y39g8b.7, daf-2;asm-2, daf-2;nhr-62* double mutants*,* the mutations of *y39g8b.7, asm-2* and *nhr-62* could partially reverted the lifespan extension of *daf-2;set-21* in triple mutants. Actually, none of the ten selected targeted genes could shorten the lifespan of *daf-2* animals but seven of them specifically shorten the lifespan in triple mutants.

2) The text needs to be rewritten to reflect that this is all only occurring in daf-2 mutant worms and not in WT worms. This rewriting needs to start with the title ("…limits the lifespan of daf-2 mutant *C. elegans*". ) and work its way thru the intro, results and discussion (where the authors claim H3K9me regulators should be considered for lifespan regulation). Need to include explicit writing that deletion of all these putative enzymes has only modest effects on H3K9me and lifespan in the WT context and their role is only revealed in the daf-2 mutants. Need to speculate about what that is. What makes the daf-2 mutation special to reveal a regulatory role for these genes?

Thanks very much for the suggestions.

We have revised the manuscript as suggested, stating that these SET proteins and H3K9me2 may preferentially limit the lifespan of *daf-2* mutant worms.

The depletion of SET proteins or the inhibition of worm H3K9me1/2 methyltransferases by G9a inhibitor significantly extended the lifespan only in *daf-2* mutant worms, but has modest effects on N2 animals. Consistently, the long-lived *set-6, set-19, set-20, set-21, set-32* and *set-33* mutants only significantly decreased global H3K9me1/2 levels in *daf-2* mutants but not in N2 animals. These results together suggested that DAF-2 limits the effects of these SET proteins.

It is unclear how and why these SET proteins only limit lifespan in *daf-2* animals. There are three possibilities. First, DAF-2 may have profound effect on the chromatin accessibility. In *daf-2* mutants, the chromatin may be more compact and prohibit the activity of the SET proteins to conduct H3K9me2 modification. Second, *daf-2* mutation may sensitize the animals by increasing the nuclear accumulated DAF-16. Although it is unclear whether DAF-16 preferentially recognizes H3K9me2 or H3K9me3, we observed a much stronger enrichment of DAF-16 in the nucleus in *daf-2;set-21* than in *daf-2* animals, which suggested that H3K9me2 may prohibit the accumulation of DAF-16 in the nucleus. Third, DAF-2 may have other indirect effect on the SET proteins. Further chromatin-immunoprecipitation experiments can be used to test whether *daf-2* mutation could affect the association of the SET proteins with chromatin. The ATAC-seq technology may allow us to investigate whether *daf-2* mutation have changed chromosome accessibility to facilitate the activity of SET proteins. In addition, identification of the substrate of SET proteins may help to elucidate their mechanism and functions.

In *daf-2* animals, DAF-16 modestly accumulated in the nucleus, as shown by the newly generated single-copy GFP-tagged DAF-16 transgene. Yet in *daf-2;set-21* double mutants, DAF-16 was strongly enriched in the nucleus. The depletion of DAF-16 completely blocked the synergistic lifespan extension in *daf-2;set-21* animals. The mRNA-seq experiments revealed that the mRNA levels of DAF-16 class I genes were elevated in the long-lived *daf-2;set-19*, *daf-2;set-21*, and *daf-2;set-32* double mutants compared to those in control *daf-2* and short-lived *daf-2;set-25* mutants. Together, these data suggested that the activity to DAF-16 to class I genes not only depends on the absence of insulin pathway, but also depends on the depletion of these SET proteins and H3K9me2. The single-copy GFP-tagged DAF-16 transgene may allow us to re-visit the mechanism of lifespan regulation pathways.

We have revised the Discussion section to include the above information (lines 435-468).

Reviewer #2 (Recommendations for the authors):Upon resubmission, this paper is greatly improved – readers now have the evidence needed to evaluate the paper's main claim that H3K9 HMTs synergistically extend lifespan in daf-2 mutants, in part via DAF-16, but also via. The additional experiments, along with a more tempered approach to discussing their findings, significantly strengthen the paper.However, several of our concerns were not addressed by these changes, nor in the rebuttal. For example, Reviewer 2 asked that some of the key lifespan experiments be replicated at least once and also performed blinded – there are several figures where only a single replicate is shown. Key controls are still missing, like in Figure 7, where Reviewer 1 had asked for appropriate controls to interpret the epistasis experiments for example, set-21 single mutants and daf-2; lys-7 double mutants. The authors' rebuttal mentioned generating new alleles for some genes, but providing appropriate single mutant and double mutant controls should require no new strains to be built (since the strains likely already exist if the authors generated the triple mutant).

Thanks very much for the comments.

We conducted new experiments to compare the lifespan of *daf-2*, *daf-2;y39g8b.7, daf-2;set-21* double mutants *vs daf-2;set-21;y39g8b.7* triple mutant and etc… (new Figure 7B). Although we did not find significant lifespan difference between *daf-2* vs *daf-2;y39g8b.7, daf-2;asm-2, daf-2;nhr-62,* the mutations of *y39g8b.7, asm-2* and *nhr-62* could partially reverted the lifespan extension of *daf-2;set-21* in triple mutants. Actually, none of the ten selected targeted genes could shorten the lifespan of *daf-2* animals but seven of them specifically shorten the lifespan in triple mutants.

We also included new blinded replicates for Figures 1A, 4A-B and 7B. The conclusions are not changed.

The extended analysis of the mRNA-seq is better (and opens up an interesting line of questioning for future studies!). However, the authors failed to address most of Reviewer 2's Major Point 3 about their transcriptomic analysis, nor did they include further details about how the analysis was done. The methods for the mRNA-seq will need to be more detailed. What is their unbiased approach? What is the differential expression shown in the Figure 5? For example, I had previously asked: "Does the color refer to upregulation in comparison to a wild-type control? Why not compare misregulation to a daf-2 single mutant instead?" and the answer is still not clear. Furthermore, if Class I genes are defined as being induced in daf-2 mutants, why are they downregulated in Figure 5E? And if Class II genes are defined as being not induced in daf-2 mutants, why are they upregulated in Figure 5F? I may be misunderstanding the gene classifications, but it is a demonstration of how the text is unclear. Overall, I found the new transcriptomic analyses are compelling, but more care needs to be taken to fully explain what was done.

Thanks very much for the comments.

We have analyzed the mRNA-seq data in two ways. The first one is comparing the expression levels of reported DAF-16 target genes in *daf-2* and *daf-2;daf-25* animals and in the long-lived *daf-2;set-19*, *daf-2,set-21*, *daf-2;set-32* double mutants (Figure 5E and Figure 5—figure supplement 2A). The second method is directly comparing the gene expression in short-lived *daf-2;daf-25* animals and the long-lived *daf-2;set-19*, *daf-2,set-21*, *daf-2;set-32* double mutants to the gene expression in *daf-2* animals (Figure 6A and Figure 6—figure supplement 1A). This method identified 49 co-upregulated genes and 11 co-downregulated genes that are specifically enriched in the long-lived double mutants but not in control *daf-2* and *daf-2;daf-25* animals. We then analyzed a number of known transcription factors for their binding to the co-regulated targeted genes.

Figure 5E showed heatmap of the standardized FPKM of reported top 50 DAF-16 Class I and Class II genes by mRNA-seq in the indicated animals. Figure 5—figure supplement 2A showed heatmap of the standardized FPKM for reported 1576 DAF-16 Class I and 1653 Class II genes by mRNA-seq in the indicated animals. Statistical analysis was performed to obtain the expression levels of each gene in each strain and the average of the expression levels in the five strains. The gene expression levels in a single strain were compared with the average of the five strains, and the ratio obtained was processed by log2. The resulting value > 0 means the expression level of the gene in the indicated strain is higher than the average expression levels of this gene in the five strains, as shown from yellow to red. The resulting value < 0 means the expression level of the gene in the indicated strain is lower than the average expression levels of this gene in the five strains, as shown in blue. The expression levels are indicated by the color bar. Most of the DAF-16 Class I genes are yellow and red in the long-lived *daf-2;set-19*, *daf-2,set-21*, *daf-2;set-32* double mutants, but blue in *daf-2* and *daf-2;daf-25.* The mRNA levels of DAF-16 Class I genes are consistently elevated in long-lived *daf-2;set-19, daf-2;set-21* and *daf-2;set-32* worms, but not in the control *daf-2* and *daf-2;set-25* animals.

We have revised the figure legends to describe our data analysis in more detail (lines 1118-1127, 1225-1234).